# A Comprehensive Framework for Transportation Infrastructure Digitalization: TJYRoad-Net for Enhanced Point Cloud Segmentation

**DOI:** 10.3390/s24227222

**Published:** 2024-11-12

**Authors:** Zhen Yang, Mingxuan Wang, Shikun Xie

**Affiliations:** Key Laboratory of Road and Traffic Engineering of the Ministry of Education, Tongji University, 4800 Caoan Rd, Shanghai 201804, China; 2031386@tongji.edu.cn (M.W.); xieshikun521@tongji.edu.cn (S.X.)

**Keywords:** transportation infrastructure digitization, semantic segmentation of road point cloud, lightweight road infrastructure reconstruction

## Abstract

This research introduces a cutting-edge approach to traffic infrastructure digitization, integrating UAV oblique photography with LiDAR point clouds for high-precision, lightweight 3D road modeling. The proposed method addresses the challenge of accurately capturing the current state of infrastructure while minimizing redundancy and optimizing computational efficiency. A key innovation is the development of the TJYRoad-Net model, which achieves over 85% mIoU segmentation accuracy by including a traffic feature computing (TFC) module composed of three critical components: the Regional Coordinate Encoder (RCE), the Context-Aware Aggregation Unit (CAU), and the Hierarchical Expansion Block. Comparative analysis segments the point clouds into road and non-road categories, achieving centimeter-level registration accuracy with RANSAC and ICP. Two lightweight surface reconstruction techniques are implemented: (1) algorithmic reconstruction, which delivers a 6.3 mm elevation error at 95% confidence in complex intersections, and (2) template matching, which replaces road markings, poles, and vegetation using bounding boxes. These methods ensure accurate results with minimal memory overhead. The optimized 3D models have been successfully applied in driving simulation and traffic flow analysis, providing a practical and scalable solution for real-world infrastructure modeling and analysis. These applications demonstrate the versatility and efficiency of the proposed methods in modern traffic system simulations.

## 1. Introduction

With the advancements in 3D laser scanning systems and unmanned aerial vehicle (UAV) photogrammetry technologies [1], these efficient, accurate, and large-scale data acquisition methods have been widely applied in various areas, including national key engineering construction, emergency responses to natural disasters, land surveillance, and smart city development. Three-dimensional laser scanning and UAV aerial photography technologies enable the creation of high-quality urban 3D GIS models and provide strong support for smart city construction when combined with BIM technology. GIS-based urban road scene models can serve smart city development, while three-dimensional models of mountainous road slopes can be used to analyze changes in terrain and geomorphology, as well as for disaster warning and monitoring. The digitalization of transportation infrastructure has provided strong technical support for intelligent transportation, autonomous driving, and digital twins. However, the current level of digitalization in transportation infrastructure is not high due to various factors such as equipment and technology limitations. There is further research needed for developing systematic, streamlined, and high-precision operational approaches for the rapid digitalization of infrastructure, taking into account the unique characteristics of the infrastructure itself.

In the field of point cloud semantic segmentation, the segmentation of road infrastructure from large-scale point cloud data still presents certain challenges due to the inherent complexity and diversity of traffic scenes. In the existing research, Dinesh et al. [2] employed histogram analysis to process depth data line by line and extract point cloud data that form the road surface. Qiu et al. [3] utilized the RANSAC method to obtain a road plane and solve the road geometry based on prior knowledge of road width and continuity. Region-growing algorithms primarily utilize the normal vectors and curvature of point clouds to group together similar points and form regions, making them one of the commonly used algorithms for point cloud semantic segmentation. Rabbani et al. [4] proposed an alternative method, which involves selecting the residuals of seed points’ planes and then using estimated point normals and residuals to select growing regions. Among the model fitting-based methods, a commonly used algorithm is the RANSAC method introduced by Bolles et al. [5]. The RANSAC method [6] iteratively estimates mathematical model parameters from observed data containing outliers and randomly selects a minimal subset of data points to extract shape primitives, constructing candidate shapes. The RANSAC algorithm only utilizes the spatial information of point clouds and is greatly influenced by density and accuracy. On the other hand, clustering-based segmentation algorithms use different features for clustering depending on the adaptive scene, requiring more prior knowledge. With the advancement of computer processing power and the increasing adoption of machine learning and deep learning techniques, a range of novel approaches for point cloud semantic segmentation have emerged. Initially, point cloud semantic segmentation was primarily based on machine learning theory. One approach is point-based classification, exemplified by Hackel et al. [7], who created a scale pyramid based on density, generated a 144-dimensional feature vector, and trained a classifier for direct classification of outdoor scene point clouds. Weinmann et al. [8] performed nearest-neighbor analysis and extracted features for classification purposes. Another approach is segmentation-based classification, which involves segmenting the point cloud data and subsequently classifying the segmented point clouds. Binin et al. [9] established point adjacency relationships based on normal vectors and utilized support vector machines for classification. The field of image segmentation has seen remarkable progress with the continuous advancements in deep learning. Consequently, certain ideas and methods have been adapted and applied to point cloud segmentation. Currently, deep learning networks for point clouds can be broadly categorized into three groups:
(a)Deep learning networks utilizing 2D projection: Lawin et al. [10] and other researchers align 3D laser point clouds with 2D images, projecting the 3D point clouds onto 2D pixels. They then employ convolutional algorithms from the image domain for semantic segmentation. Boulch et al. [11] integrate multi-view projection with pretrained models for point cloud classification. However, these methods in this category encounter limitations due to the loss of spatial information during the projection process, affecting their overall effectiveness.(b)Convolutional neural networks utilizing 3D voxelization: In order to address the spatial information loss in projection methods, Dniel et al. [12] propose the VoxNet model, which voxelizes point clouds and combines them with 3D convolutional neural networks. Gernot et al. [13] introduce the OctNet network model, employing an unbalanced octree and 3D convolutional neural networks. These methods tackle the spatial structural patterns of point clouds, effectively addressing issues of permutation and rotation invariance. However, they solely focus on the spatial information of point clouds [14], resulting in some information wastage and potential for further optimization.(c)Individual point-based network models: To address large-scale point cloud semantic segmentation, Hu et al. [15] and Fan et al. [16] propose the RandLA-Net and SCF-Net, respectively. These networks utilize random sampling methods during the encoding process to reduce point cloud density and optimize memory usage. They outperform PointNet [17] and PointNet++ [18] in segmenting large-scale point clouds. However, current deep learning-based point cloud segmentation in road scenes mainly focuses on extracting individual objects. Tao et al. [19] combine learning-based (PointNet++) and non-learning-based (P4UCC) methods for large-scale segmentation of road infrastructure point cloud models. The results indicate that classifying smaller targets such as cars and poles remains challenging due to the limited dataset size [20].

In the field of multi-source point cloud fusion, registration based on the inherent features of the point clouds is typically required. To enhance the performance of the traditional iterative closest point (ICP) algorithm in unstructured environments, Yao et al. [21] proposed an improved iterative closest point algorithm that utilizes the similarity-of-curvature features in point clouds. They aimed to address the limitations of the traditional ICP. Ren et al. [22] introduced a hue-based color point cloud registration algorithm that demonstrates robustness under various lighting conditions. This algorithm incorporates color information to improve registration accuracy compared to the ICP. Sui et al. [23] proposed a deep learning model based on a three-dimensional Siamese convolutional neural network to tackle the point cloud registration problem caused by data noise or incompleteness. For registering large point clouds with low accuracy and slow speed, Xu et al. [24] presented an improved iterative closest point algorithm that combines the RANSAC and the ISS. This algorithm achieves a faster registration speed compared to other methods.

Point cloud surface reconstruction can be categorized into three types: parameter-based surface reconstruction [25], geometry-based triangulation, and implicit surface reconstruction methods. Hoppe et al. [26] proposed using B-spline surfaces to construct the corresponding surface normals of the object to be reconstructed. Kazhdan et al. [27] introduced the Poisson surface reconstruction algorithm, which takes a global approach to point clouds and offers better reconstruction flexibility and accuracy compared to radial-basis functions. Xu et al. [28] developed a fully automated method for reconstructing large-scale scenes using multi-site point clouds. Gao et al. [29] achieved the fusion of ground and aerial point clouds by linking matched point pairs from the ground to the air with their original trajectories and bundle adjustment. Li et al. [30] used a multi-view projection method to detect holes in TLS point clouds and extracted corresponding data from the registered dense matching point clouds to fill the gaps. Thus, existing scholars have achieved the fusion of heterogeneous point cloud data through the registration of disparate point clouds.

Current 3D reconstruction and semantic segmentation technologies for transportation infrastructure face challenges in balancing lightweight design, high accuracy, and realistic representation. Existing semantic segmentation networks, though effective in general environments, often struggle in large-scale traffic scenes where they fail to capture the intrinsic characteristics of infrastructure elements accurately. These networks typically lack the capacity to recognize and learn features unique to transportation point clouds, such as the continuity of road markings, the planar nature of road surfaces, and the structural characteristics of traffic poles and signage. Additionally, the absence of a mature, integrated workflow for processing and reconstructing complex scenes limits the applicability of these methods in real-world traffic environments. Addressing these gaps, TJYRoad-Net, proposed in this paper, is a semantic segmentation network specifically developed for large-scale traffic point clouds, demonstrating strong performance in segmenting extensive traffic scenes. Designed to effectively capture and leverage critical infrastructure features, TJYRoad-Net lays a solid foundation for accurate 3D reconstruction tailored to various infrastructure elements, thus supporting realistic modeling and further enabling advanced applications in transportation infrastructure digitization.

The integration of UAV oblique photogrammetry and LiDAR data is crucial due to the complementary strengths and limitations of each data source. UAV photogrammetry offers extensive coverage, capturing entire buildings, complete trees, and other features that might be partially absent in LiDAR scans. However, photogrammetry is limited by a lower accuracy and can exhibit deformation in reconstructing trees, poles, and other small structures, with noticeable noise on road surfaces. Conversely, LiDAR provides high-precision data, accurately capturing detailed features near road surfaces but is restricted in range, often scanning only one side of buildings. Combining these two sources allows us to leverage the high accuracy of LiDAR with the wide coverage of UAV photogrammetry, producing a highly accurate and realistic model that is both comprehensive and true to real-world dimensions.

This study combines 3D laser scanning technology and multi-angle aerial photography using UAVs to collect data on transportation infrastructure. By segmenting the image-derived and laser-derived point clouds separately, followed by targeted fusion for different types of transportation infrastructure, this approach achieves higher accuracy compared to direct large-scale scene fusion. It integrates laser point clouds and image matching point cloud data to digitize infrastructure. By addressing technical challenges related to point cloud acquisition, segmentation, fusion, and surface reconstruction specific to transportation infrastructure characteristics, this study aims to generate lightweight yet realistic models tailored to various transportation application requirements. These models can be utilized for tasks such as driving simulations and traffic simulations, enhancing the accuracy and efficiency of digitalization in infrastructure and facilitating future research in a more convenient and efficient manner. This study aims to contribute to the comprehensive digitalization of transportation infrastructure. The overall study follows the technical roadmap depicted in Figure 1.

## 2. Materials and Methods

### 2.1. Acquisition of Point Clouds from UAV-Based Oblique Imagery

Using camera-enabled UAV and 3D laser scanners separately, tasks are conducted within a certain range. The five-flight route mode (comprising one nadir flight and four oblique flights at 45-degree angles in the four cardinal directions) is utilized to capture aerial images of road infrastructure from multiple angles and collect image data. The Multi-View Stereo (MVS) and Structure From Motion (SFM) algorithms are employed to obtain complete and dense point clouds of the road infrastructure. Additionally, a vehicle-mounted laser scanner is used to acquire laser point clouds of the road infrastructure that have high overlap with the image data area.

The way to generate image point clouds is to carry out motion recovery structure SFM (Structure From Motion) for tilted images. Estimating the pose information of the tilted image and creating the object structure information of the spatial scene determines the spatial and geometric relationship of the target according to the dynamic movement of the camera, thus transforming the two-dimensional image information into three-dimensional spatial depth information and completing the three-dimensional reconstruction of the road infrastructure point cloud. DJI Terra can directly generate 3D models using oblique photogrammetry; however, for large-scale transportation scenes, it often produces gaps and streaking artifacts in model details close to the ground, especially around infrastructure like signs and trees (see https://www.dji.com/dji-terra, accessed on 15 June 2023).

This paper uses the DJI Matrix M300RTK and Zenmuse P1 gimbal camera for data acquisition, as shown in Figure 2; the technical specifications are shown in Table 1.

Taking the Jiading Campus of Tongji University in Shanghai as the photogrammetry area, the survey area includes three main campus roads—Tongjia Road, Tongyi Road, and Jiasan Road—as well as the façades of five buildings—Tongda Museum, Kaiwu Museum, Ningyuan Museum, Decai Museum, and Zhixin Museum—and the network RTK is turned on throughout the photography process, and the isochronous interval photography mode is adopted; a total of 194 orthophotos and 623 oblique images are obtained.

Once all the feature points from the road infrastructure images are extracted, a pairwise matching of these feature points is required. However, using the brute-force matching algorithm in this scenario is computationally expensive due to the large amount of feature information and the high number of feature points in the traffic scene images. Instead, the approximate nearest-neighbor feature matching (ANN feature matching) method is more suitable for situations with a large number of matching points, as it reduces the time complexity and computational load. In this study, the bidirectional ANN matching method is employed to match the feature points in the traffic scenes. The bidirectional ANN algorithm improves the matching accuracy with a slight increase in matching time. The steps of the bidirectional ANN algorithm are outlined below:

A feature point in image $P_{1}$ corresponding to the feature point $s_{2}$ with the smallest distance in image $P_{2}$ is obtained by the FLANN [31] algorithm (FLANN is designed to quickly perform nearest-neighbor searches in high-dimensional spaces, which makes it suitable for large-scale image-matching tasks by efficiently constructing an initial set of matching point pairs) to construct the initial matching point pair ($s_{1}$,$s_{2}$), and then, the smallest distance is taken out of all the matching point pairs, and the metric parametric D is set, and if the smallest distance of the matching point pair is less than D, then $s_{2}$ is regarded as a candidate matching point for $s_{1}$. Otherwise, the next feature point matching of image $P_{1}$ is performed, and $s_{1}$ is rejected, so as to obtain the matching point pairs of image $P_{1}$ and image $P_{2}$.Similarly, the feature points of image $P_{2}$ are processed similarly to the previous step, and the matching point pairs of image $P_{2}$ and image $P_{1}$ are obtained. The candidate point $s_{3}$ in image $P_{1}$ that matches with $s_{2}$ of image $P_{2}$ is compared and judged, and if $s_{3}$ is the same point as $s_{1}$, the matching is successful; otherwise, it is rejected.This process is repeated for all feature points in both images, resulting in a robust set of matched points that have been verified from both directions. The bidirectional matching process effectively filters out ambiguous or incorrect matches, improving the overall accuracy of the feature matching process and ensuring more reliable alignment of the infrastructure features.

#### Camera Motion Estimation and Point Cloud Generation

After obtaining a set of matched traffic scene feature points, the camera motion between two frames can be recovered using the correspondence of these 2D image points. This involves calculating the rotation parameter R translation parameter t between two frames of traffic scene images, denoted as $P_{1}$ and $P_{2}$. Assuming the camera centers of the two frames are $O_{1}$ and $O_{2}$, respectively, and that a feature point $s_{1}$ in image $P_{1}$ corresponds to a feature point $s_{2}$ in image $P_{2}$, under the condition of correct feature point matching, these two points represent the projections on two imaging planes of the same traffic scene. According to the principle of epipolar constraint, their relationship can be expressed as follows:(1)$$x_{2}S_{2}^{T}K^{-T}EK^{-1}S_{1}=S_{2}^{T}FS_{1}=0$$

The motion estimation of the UAV camera can be simplified into the following two steps:The essential matrix E and the fundamental matrix F are calculated using the pixel positions of the matched feature points to estimate the geometric relationship between the two frames.By utilizing the essential matrix E and fundamental matrix F, the camera motion, represented by the rotation parameter R and translation parameter t, can be determined. These matrices provide crucial information about the geometric transformation between the two camera frames and enable accurate estimation of the camera’s position and orientation changes.

In this paper, the seven-point method is employed to estimate the essential matrix E. The AC-RANSAC method is utilized for outlier rejection in the feature point matches. AC-RANSAC is a variant of RANSAC that estimates stable model parameters by computing residuals and then evaluates the reliability of the model using the Number of False Alarms (NFA) evaluation function. Compared to RANSAC, AC-RANSAC requires fewer parameters and enables direct and objective comparison of model differences, thereby obtaining the best model.

Once the elements of the essential matrix E are obtained, they are subjected to singular value decomposition (SVD), and by substituting them into any point, R and t can be calculated. Triangulation is employed to derive the spatial positions and depth information of the feature points, allowing the mapping of the feature points to the 3D space and obtaining a sparse point cloud representation of the traffic scene.

The sparse reconstruction results in optimized UAV camera poses and a sparse point cloud that provides an initial representation of the overall shape of the road infrastructure. However, it may not accurately capture surface details such as cracks and potholes on the road. Consequently, the road infrastructure is reconstructed in a dense manner by employing depth map calculation and fusion techniques. This process involves estimating the depth information for each pixel in the scene and combining multiple depth maps to generate a dense representation of the traffic scene. The resulting dense point cloud accurately captures the geometric details of the road infrastructure. In Figure 3, the complete process of generating a large-scale point cloud of transportation infrastructure using oblique photography is described. 

### 2.2. Acquisition of Point Clouds Based on Vehicle-Mounted LiDAR Scanning

Data collection experiments were conducted using the IP-S2 mobile laser scanning system developed by Topcon, Japan. The hardware components of the IP-S2 system consist of three SICK laser scanners, six LADYBUG panoramic CCD cameras, dual-frequency GPS GLONASS positioning receivers, an inertial navigation unit, and two wheel encoders. The laser scanners have a measurement accuracy of up to 35 mm and a range of up to 30 m. The cameras have a resolution of 1600 × 1200 pixels. The dynamic measurement accuracy of the GNSS unit is within the range of 10 mm to 15 mm.

Data collection was conducted using the IP-S2 mobile laser scanning system developed by Topcon, Japan. The technical specifications of the system are summarized in Table 2.

A selected section of Tongjia Road and Jiasan Road was chosen as the experimental area for data collection, and a path planning was performed. The center points of manhole covers and the corners of road markings were used as control points for calibration. The data collection was conducted at a speed of approximately 40 km/h, with efforts made to maintain a constant speed and drive along the center of the lane to ensure a certain degree of overlap with the UAV survey area.

After completing the field data collection, encrypted data from each component can be obtained. This includes the following:Laser point cloud data stored in the order of 3D spatial coordinates and reflection intensity, with the file extension .ips;Three-hundred-and-sixty-degree panoramic images generated by the LADYBUG cameras, with the file extension .pgr;Vehicle attitude and positioning data, with the file extension .ips.

These data are processed using Geoclean software, version 1.2. The processing steps include vehicle body masking, GNSS and vehicle attitude information processing, panoramic image generation, coordinate generation, and point cloud coloring. Finally, the point cloud representation of the road infrastructure is obtained, as shown in Figure 4.

### 2.3. Semantic Segmentation of Road Infrastructure Point Cloud

The road infrastructure point cloud is subjected to semantic segmentation using a deep learning approach. Based on the requirements analysis, the network architecture suitable for road traffic point cloud scenes needs to have the following characteristics:Lightweight grid design: The network should be computationally efficient, have low memory usage, and be capable of directly processing large batches of three-dimensional point clouds of the road infrastructure without the need for voxelization, scene segmentation, or other unnecessary pre-processing or post-processing operations.Effective local scene feature learning module: It should incorporate a progressive expansion of the receptive field to learn the complex and diverse geometric structures of road traffic scenes.Efficient and memory-friendly point cloud downsampling method: Continuous downsampling of the point cloud should be employed to ensure that the network can adapt to the available GPU memory and computational power.

Based on the aforementioned requirements, this paper proposed the TJYRoad-Net network to perform segmentation on both image-based point clouds and laser-based point clouds.

#### 2.3.1. TJYRoad-Net

The TJYRoad-Net framework, specifically designed for large-scale road traffic scenes, introduces a novel architecture to handle the challenge of semantic segmentation in complex and high-density point cloud environments. It improves segmentation performance for large-scale road infrastructure datasets, particularly those generated from oblique drone imagery. The structure of the network is shown in Figure 5.

In Figure 5a, the input point cloud has a point count and feature dimension of *N* and *X*, respectively, with the feature dimension then expanded to 8. TJYRoad-Net employs an encoder–decoder architecture with skip connections, similar to U-Net. It begins with a fully connected layer that processes the input point clouds, expanding the feature dimensions from *X* to 8, where *N* denotes the number of input points. In the subsequent sections of the figure, the network extracts point-wise features through four downsampling and four upsampling layers. In the final stage, three fully connected layers, along with a Dropout layer, generate the semantic predictions for each point.

During the encoding phase, traffic feature computing (TFC) modules and random sampling (RS) modules progressively reduce the point cloud size to 1/4, 1/16, 1/64, and 1/256 of its original size, while increasing the feature dimensions to 32, 128, 256, and 512, retaining rich feature information. In the decoding phase, instead of trilinear interpolation, the K-nearest-neighbors (KNN) algorithm is employed to find neighboring points, and upsampling is performed using nearest-neighbor interpolation (US). Skip connections are utilized to combine low-level features from the encoding stage with high-level features in the decoding stage, helping to minimize information loss. Finally, a shared MLP is used to reduce the feature dimensionality before output. After encoding and decoding, the 8-dimensional features are mapped to specific classes via fully connected layers. The TFC module is the critical component of the TJYRoad-Net network. Figure 5b,c depict its structure.

The TFC module is composed of three sub-modules: the Regional Coordinate Encoder (RCE), Context-Aware Aggregation Unit, and Hierarchical Expansion Block. By incorporating the normal vector information into spatial local encoding, the RCE is optimized for understanding structural characteristics in large-scale road traffic scenes.

(a)Enhanced RCE module: This sub-module’s main function is to encode the spatial information of the original point cloud at multiple scales, allowing the network to capture spatial structural details within each point’s local neighborhood in traffic scenes. By integrating the multi-scale neighborhood sampler (MSNS), the module samples neighboring points at different scales to better capture varying geometric structures. Specifically, for a given point $P_{i}$, MSNS performs neighborhood sampling at multiple levels using k-nearest neighbors (KNN) with varying values of $k_{m}$ for each scale m. This approach is inspired by the hierarchical feature learning in PointNet++ [18], where multi-scale grouping (MSG) is used to capture both fine- and large-scale features within point clouds:

(2)$$N_{i}^{k_{m}}=\left\{P_{j}|j\in KNN\left(P_{i},k_{m}\right)\right\}$$
where $N_{i}^{k_{m}}$ denotes the neighbors of point $P_{i}$ at the *m*-*th* scale. This multi-scale sampling enables the model to capture both fine-grained and large-scale structures.

The dynamic weighting module (DWM) further enhances the encoding process by dynamically adjusting the importance of each sampled neighbor based on its spatial relationships with the central point. This mechanism is inspired by the self-attention mechanism commonly used in models like the Transformer [32], which allows the network to assign varying weights to features based on their relevance. In our case, the weight $w_{ij}$ assigned to a neighboring point $P_{j}$ of $P_{i}$ is computed as a function of their relative spatial position and distance: (3)$$w_{ij}=softmax\left(f\left(\left\|P_{i}-P_{j}\right\|,∠\left(N_{P},P_{j}-P_{i}\right)\right)\right)$$
where $\left\|P_{i}-P_{j}\right\|$ represents the Euclidean distance between points $P_{i}$ and $P_{j}$, and $∠\left(N_{P},P_{j}-P_{i}\right)$ denotes the angle between the normal vector $N_{P}$ of point $P_{i}$ and the vector from $P_{i}$ to $P_{j}$. The dynamic weighting helps the network prioritize more significant neighbors based on their geometric relationships.

The spatial information encoded by the RCE module includes the following:

Three-dimensional coordinates of point P, denoted as $P_{i}^{K}$;Three-dimensional coordinates of neighboring points, denoted as $P_{i}-P_{i}^{K}$;Spatial relationships between point P and its neighbors Euclidean distance between point P and neighboring points, denoted as $\left\|P_{i}-P_{i}^{K}\right\|$;Normal vector information of point $P_{i}$, denoted as $N_{P}$.

Finally, the Multi-Layer Perceptron (MLP) encodes these spatial and geometric features. The resulting feature $r_{i}^{K}$ for each point $P_{i}$ is computed as follows:(4)$$r_{i}^{K}=MLP\left(P_{i}⊕P_{i}^{K}⊕\left(P_{i}-P_{i}^{K}\right)⊕\left\|P_{i}-P_{i}^{K}\right\|⊕N_{P_{i}}⊕w_{ij}\right)$$

The enhanced RCE module significantly boosts the model’s segmentation performance, primarily through the contributions of the MSNS and the DWM:

MSNS: MSNS allows the model to capture both fine-grained and large-scale structural features by sampling neighboring points at multiple scales. This enables the network to adaptively capture diverse geometric structures, such as the continuity of road markings and the planar nature of road surfaces, which are crucial in traffic scenes. By extracting neighborhood features at different levels, MSNS enhances the model’s ability to recognize various infrastructure elements more accurately.DWM: The DWM dynamically assigns importance to neighboring points based on their spatial relationship to the central point. This adaptive weighting emphasizes critical local features, such as traffic poles and signage, while downplaying less relevant points. Inspired by attention mechanisms, DWM improves the model’s focus on essential features, leading to higher accuracy in segmenting complex traffic environments.

Together, MSNS and DWM enrich the RCE module’s feature encoding capabilities, resulting in more precise and reliable segmentation of large-scale traffic scenes.

(b)Context-Aware Aggregation Unit: This unit aggregates features from the neighboring points produced by the RCE. Rather than relying on conventional max-pooling or average-pooling, the CAU employs a multi-head self-attention (MHSA) mechanism. MHSA enables the model to capture different aspects of local features from multiple perspectives by using multiple attention heads. Each attention head focuses on a distinct subspace of the features, allowing for a more comprehensive aggregation. For a given point $P_{i}$, multiple attention heads h∈{1,2,…,H} compute independent attention scores $\overset{\wedge }{C_{i}^{h}}$ for each neighboring point $P_{j}$ at scale *k*:

(5)$$\overset{\wedge }{C_{i}^{h}}=\{\overset{\wedge }{c_{i}^{h,1}},\overset{\wedge }{c_{i}^{h,2}},\cdots ,\overset{\wedge }{c_{i}^{h,k}},\cdots ,\overset{\wedge }{c_{i}^{h,K}}\}$$
where $\overset{\wedge }{c_{i}^{h,k}}$ represents the attention score assigned to the *k*-th neighbor by the *h*-th attention head. Each attention head independently aggregates the local features, and the final output is obtained by concatenating the results from all attention heads and passing them through a linear transformation.

To further enhance the aggregation, a hierarchical attention mechanism is introduced. In this mechanism, local attention is first applied within each head to capture short-range relationships, followed by a global attention that aggregates the results of the heads to capture long-range dependencies across the entire neighborhood. This hierarchical attention setup ensures that both local and global spatial relationships are preserved during feature aggregation.

The final aggregated feature $s_{i}^{k}$ for each point $P_{i}$ is computed as a weighted sum of the neighboring point features, considering both the attention scores from the heads and the global attention weights:(6)$$s_{i}^{k}=\sum_{h=1}^{H}g\left(\overset{\wedge }{c_{i}^{h.k}},W_{h}\right)$$
where $W_{h}$ are the learnable weights for the *h*-th attention head. The global attention layer combines the outputs from each head to form the final aggregated feature. This combined multi-head self-attention and hierarchical attention approach is inspired by similar techniques used in models such as RandLA-Net [15], where attention-based mechanisms are applied to enhance feature aggregation in large-scale point clouds.

The Context-Aware Aggregation Unit further enhances model performance by incorporating MHSA and a hierarchical attention approach. The MHSA mechanism enables the model to capture multiple representations of local features from different perspectives by utilizing multiple attention heads. Each head independently captures unique aspects of the input, providing a richer and more nuanced understanding of complex traffic scene elements, such as the varied textures and shapes of different infrastructure components. Meanwhile, the hierarchical attention approach balances local and global contextual information. By creating separate channels for local and global attention, the model is able to maintain a broader view of the scene structure while preserving important localized details. This dual focus allows the network to more accurately differentiate between nearby objects with similar features, such as road markings and building façades. Together, these attention mechanisms significantly boost the segmentation precision of TJYRoad-Net, particularly in large-scale and densely structured traffic environments.

(c)Hierarchical Expansion Block: This sub-module is designed to expand the receptive field of each point. Downsampling is crucial for processing large-scale road traffic point cloud scenes, as random sampling reduces time and space complexity. However, this can lead to information loss at certain critical points. Therefore, expanding the receptive field for each point becomes essential. To address this, the Hierarchical Expansion Block stacks multiple RCEs and Context-Aware Aggregation Units (CAUs) with skip connections, forming a hierarchical structure that enhances feature aggregation and preserves important spatial information.

In the final CAU layer of this block, dilated convolutions with dilation rates of 2, 4, and 6 are applied to further increase the receptive field without increasing the number of neighboring points *K*. These dilated convolutions allow the network to capture larger-scale spatial dependencies across the point cloud, making it more efficient than simply increasing *K*. By introducing multiple dilation rates, the network can simultaneously capture fine-grained and large-scale features, helping to mitigate the loss of information caused by downsampling.

The block includes shared MLP layers before and after processing, merging the output features with those from the input point cloud. This final aggregation combines the information from both the hierarchical structure and the expanded receptive field, resulting in a more comprehensive and informative representation of the road traffic scene.

#### 2.3.2. Object Segmentation Based on Transfer Learning

Conventional machine learning techniques for semantic segmentation typically demand unique data annotation for each distinct objective. Consequently, training the model needs to be performed separately for each objective, resulting in repetitive processes. This approach is time-consuming and labor-intensive. In contrast, transfer learning enables the transfer of model parameters obtained from training on existing datasets to a new model in a certain way, thereby accelerating the convergence speed and training efficiency of the new model. Additionally, training the new model while optimizing the parameters of the old model improves training and prediction accuracy, as shown in Figure 6. 

Fine-tuning, a commonly used method in transfer learning, builds on the core concept of transfer learning, which involves retraining a model that has already been pretrained on a similar dataset. In this paper, fine-tuning experiments are conducted based on the enhanced TJYRoad-Net following these steps:Train the enhanced TJYRoad-Net network model using the source domain dataset to obtain the source model for point cloud segmentation.Construct a new TJYRoad-Net model, referred to as the target model, by replicating the entire architecture and parameters of the source model, except for the output layer, which includes the encoder and decoder.Add an output layer to the target model with the number of classes specific to the target dataset and randomly initialize the parameters of this output layer.Train the target model using the target dataset. Select the decoding layers near the output layer within the hidden layers, and perform fine-tuning with a smaller learning rate while retraining the output layer.

The fine-tuning process for TJYRoad-Net is illustrated in Figure 7. This approach leverages the pretrained model’s learned features from large-scale road traffic point cloud data, while adapting to the specific requirements of the new target dataset with minimal adjustments to the network’s deeper layers, thereby maintaining efficiency and accuracy in point cloud segmentation.

To take advantage of open-source datasets containing annotated point cloud information, the Semantic3D [33] (http://www.semantic3d.net/) and SemanticKITTI [34] (http://www.semantic-kitti.org/) datasets were chosen as the source domain datasets. For the target domain, a smaller-scale annotation process was performed, resulting in the creation of a training set and a validation set. The annotated classes include lane markings, road surfaces, vegetation areas, vehicles, poles, trees, and buildings, which are assigned numbers ranging from 0 to 6. 

### 2.4. Fusion of Multi-Source Infrastructure Point Cloud

Aligning multiple point clouds in terms of spatial–temporal and reference consistency is a necessary prerequisite for extracting geometric information and conducting 3D modeling applications on road assets. The precise registration of laser point clouds and image point clouds is crucial for achieving spatial–temporal and reference unification.

#### 2.4.1. Fusion Requirements Analysis

After conducting a comparative analysis of laser scanning point clouds and image point clouds, this study concludes that laser point clouds, equipped with Global Positioning System (GPS) and inertial navigation system data, exhibit superior accuracy and stability in comparison to image point clouds. Therefore, they are suitable as a basis for precision correction during the point cloud fusion stage. On the other hand, image point clouds possess richer texture information and more comprehensive scene coverage, making them useful for compensating for the viewpoint limitations of laser point cloud data and providing additional scene details [35].

Considering the higher noise level in image point clouds compared to vehicle-mounted laser point clouds, it is suggested that the vehicle-mounted laser point clouds are treated as the target point cloud during the registration process. On the other hand, the image point clouds should be considered as the point cloud requiring registration. To achieve accurate alignment, a coarse-to-fine registration process is recommended.

#### 2.4.2. Coarse Registration

Figure 8 showcases the image point cloud and laser point cloud before the registration process takes place, where it can be observed that the two point clouds have the same scale in the coordinate space, eliminating the need for scaling operations. Only certain rotation and translation transformations are required to complete the registration process.

The RANSAC algorithm is applied in this study for the coarse registration of point clouds. The steps involved in this process are as follows:Identify at least three pairs of corresponding points between the source and target point clouds to establish an initial set of correspondences.Randomly select three pairs from the correspondence set to compute a preliminary rigid transformation matrix.Use this transformation to assess each correspondence pair by calculating distance errors, classifying pairs as inliers if the error is below a specified threshold and as outliers otherwise.Repeat the random selection and transformation computation multiple times, tracking the number of inliers for each iteration. The model yielding the maximum number of inliers is retained as the optimal transformation.The final rigid transformation matrix, representing the best fit for coarse registration, is shown below:
$$\left[R|T\right]=\left[\begin{array}{cccc} 0.062 & -0.998 & 0.007 & 372.009\\ -0.994 & -0.062 & -0.091 & 530.377\\ -0.091 & -0.012 & -0.996 & 32.770\\ 0 & 0 & 0 & 1 \end{array}\right]$$

#### 2.4.3. Fine Registration

Fine registration requires the point cloud to be in an ideal initial position and entails a significant overlap between the two point clouds. The ICP algorithm is utilized for fine registration between the laser and image point clouds in this study. Furthermore, since the point clouds of the traffic scene have been accurately segmented into different semantic labels (achieving 86% accuracy for the laser point cloud and 90% for the image point cloud), it becomes possible to employ different registration approaches based on the segmentation outcomes of distinct semantic labels.

The part of the point clouds with the highest overlap between the laser and image point clouds is the road surface, which is crucial for subsequent infrastructure 3D modeling and applications. Hence, the segmented point cloud of road surface can be registered with high precision as a separate entity. Other point clouds, such as poles, trees, and vegetation areas, have fewer data points and exhibit irregularities, which make them unsuitable for fine registration purposes. Conversely, the building point clouds demonstrate regularity and high accuracy in both the laser and image point clouds. Building point clouds exhibit unique characteristics in both the laser and image point clouds. In the laser point cloud, buildings are observed as planar façades, whereas the image point cloud represents them as complete 3D models. Leveraging this consistency, this study employs a dedicated method for extracting building façades to achieve accurate registration of non-road surface point clouds.

### 2.5. Lightweight Surface Reconstruction of Road Infrastructure

For automated surface reconstruction from point clouds, various methods exist, including the Delaunay triangulation-based algorithm, region-growing algorithms, implicit surface reconstruction methods, and deep learning-based algorithms. Among these, the Poisson surface reconstruction algorithm [27] was selected for this study due to its ability to generate smooth, continuous surfaces with minimal holes, which is well suited for road infrastructure modeling. Poisson reconstruction constructs a vector field through an integral relationship between sample points and indicator functions, capturing surface details efficiently, though the processing speed is affected by octree depth. While alternative methods, such as Greedy Projection Triangulation, offer faster reconstruction, they struggle with uneven densities and non-planar surfaces, making them less ideal for the requirements of this study.

## 3. Results

### 3.1. Experimental Setup for Point Cloud Segmentation

To compare the impact of the improved TJYRoad-Net network, direct training, and fine-tuning experiments on the overall accuracy of road infrastructure point cloud segmentation, this study conducted corresponding controlled experiments. The experiments were conducted on an Ubuntu 18.04 server with a TensorFlow 2.6.0 deep learning framework, using an RTX 3080 GPU with CUDA 11.4. Pretraining and fine-tuning experiments were performed on image point clouds and LiDAR point clouds. The network hyperparameters for the controlled experiments are shown in Table 3.

In the context of experimental groups, the first and sixth groups function as control groups. In the first group, the pre-improved TJYRoad-Net network undergoes direct training on the target domain dataset for 20 epochs, utilizing a learning rate of 0.01. Similarly, the sixth group involves the direct training of the post-improved TJYRoad-Net network on the target domain dataset for 20 epochs, employing a learning rate of 0.01. The target domain dataset itself was manually segmented by our team from small-scale image and laser point clouds to establish the final set of class categories. In contrast, the remaining groups are experimental and focus on fine-tuning the network in different ways after pretraining with the source domain data.

The network’s point cloud segmentation performance is evaluated using several metrics: overall accuracy (OA), class-wise intersection over union (IoU), and mean intersection over union (mIoU). During evaluation, TP represents the number of correctly segmented point clouds, FP represents false-positive point clouds, and FN represents false-negative point clouds.

The calculation formulas for these metrics are as follows: (7)$$\begin{array}{l} OA=\frac{1}{N}\sum_{C}TP_{i}\\ IoU=\frac{TP}{TP+FP+FN}\\ mIoU=\frac{1}{C+1}\sum_{i=0}^{C}IoU \end{array}$$

### 3.2. Experimental Setup for Laser Point Cloud

Laser point clouds have a smaller quantity compared to image point clouds and contain less spatial information. To prevent overfitting during the experiments, it is appropriate to reduce the network depth. This study conducted experiments using an eight-layer TJYRoad-Net network.

Training the source domain laser point cloud dataset took approximately 35 h, with 65 and 100 training iterations. Directly training on the target domain dataset, the TJYRoad-Net model showed faster convergence, taking 30 min with 20 training iterations. The fine-tuning experiments on the target domain took 20 min with 15 training iterations. Table 4 and Figure 9 present the experimental results for the segmentation of the laser point cloud.

### 3.3. Experimental Setup for Image Point Cloud

Image point clouds contain a larger amount of information, and their point cloud density is higher. This study conducted experiments using a ten-layer TJYRoad-Net network. Since there are limited annotated point cloud datasets specifically generated from UAV oblique images in existing publicly available datasets, and to ensure a reasonable alignment between the source domain dataset and the target domain dataset, the fine-tuning experiments in this study were based on the assumption that the dissimilarity between the two datasets should be limited. Therefore, only the Sensat Urban dataset was chosen as the source domain dataset.

Similarly, the target domain point cloud dataset, derived from UAV imagery, was annotated within a defined range. This annotated dataset was subsequently divided into the same seven classes as the target domain dataset for the laser point cloud. 

Training the UAV source domain dataset took a total of 25 h, with 100 training iterations. Directly training the model on the target domain dataset showed faster convergence, taking 40 min with 20 training iterations. In the fine-tuning experiments on the target domain, the training duration was 30 min with 15 training iterations. Table 5 and Figure 10 display the segmentation results obtained from the image point cloud experiments.

### 3.4. Analysis of Semantic Segmentation Results

In this study, we applied transfer learning and fine-tuning to each of the comparative networks, including PointNet [17], PointNet++ [18], RandLA-Net [15], SCF-Net [16], and KPConv [36], using our custom large-scale traffic scene dataset. This approach ensured that each model was optimized for the specific characteristics of our dataset, providing a fair and accurate comparison of their performance in segmenting road infrastructure and traffic-related objects.

This paper evaluates the performance of TJYRoad-Net on the traffic scene dataset used in this study. The accurate segmentation of road markings and boundaries (especially road edges) is crucial for subsequent tasks like 3D reconstruction. The results demonstrate that TJYRoad-Net is capable of producing overall accurate segmentation, particularly for critical features such as road surfaces and markings, which are essential for defining drivable areas and detecting lane boundaries.

The network shows robustness in segmenting road surfaces, where its performance remains stable across various challenging traffic scenes, even in the presence of occlusions such as vehicles or shadows cast by nearby objects. However, there are some notable misclassifications observed in certain scenarios. For instance, some trees were incorrectly segmented as buildings, and parts of buildings near the ground were occasionally mislabeled as vegetation. Additionally, certain low-height signs were classified as buildings, likely due to similarities in geometry at those specific scales.

These misclassifications, while minimal, point to areas where the network could potentially be improved by refining the local feature aggregation or enhancing the contextual information used to differentiate between objects of similar geometric profiles. Despite these challenges, TJYRoad-Net remains highly robust in road segmentation tasks, with very few errors in identifying road boundaries, lane markings, and surfaces, which are key for subsequent processes like mapping, 3D reconstruction, and vehicle navigation. The network’s ability to correctly identify these crucial features significantly aids in high-precision 3D modeling of urban environments.

The results highlight that while TJYRoad-Net has demonstrated strong performance, further optimization, such as incorporating additional attention mechanisms or enhancing multi-scale feature aggregation, could help address some of the current segmentation errors and improve the overall accuracy in more complex scenarios. The Semantic segmentation result of image point cloud is shown in Figure 11. 

The comparison of TJYRoad-Net with other leading point cloud segmentation networks highlights the key innovations that make it stand out, particularly in large-scale traffic scenes. The Comparison with state-of-the-art methods is shown in Table 6 and Figure 12. While models like PointNet and PointNet++ have known limitations in handling local geometric details and complex object structures, it is essential to focus on the comparison with more competitive models like RandLA-Net, SCF-Net, and KPConv, which also excel in large-scale segmentation tasks but have distinct differences in how they process point cloud data.

Road Markings and Surfaces: In traffic scenes, precise segmentation of road markings and road surfaces is critical for tasks like autonomous driving and 3D urban reconstruction. TJYRoad-Net achieves an IoU of 79.28% on road markings and 95.34% on road surfaces, outperforming RandLA-Net (73.43% and 90.12%) and SCF-Net (74.32% and 91.45%). The superior performance of TJYRoad-Net is largely attributed to the CAU (Context-Aware Aggregation Unit), which dynamically aggregates features, focusing on the most relevant local geometric details such as road lines and surface boundaries. RandLA-Net and SCF-Net also utilize attention mechanisms, but TJYRoad-Net benefits from the RCE (Regional Coordinate Encoder), which encodes not only the spatial position but also normal vectors. In road segmentation, where road surfaces typically have normal vectors pointing upward, the DWM within the RCE leverages this to enhance feature aggregation for better segmentation results.Buildings and Vertical Structures: In the category of buildings, TJYRoad-Net achieves an IoU of 97.22%, comparable to KPConv (96.78%). The challenge in segmenting buildings, especially façades, lies in accurately capturing their vertical geometry, as building surfaces often exhibit normal vectors parallel to the ground. The RCE module in TJYRoad-Net is highly effective in this context, utilizing these normal vectors to differentiate between building façades and other objects such as trees or vehicles. KPConv also performs well here, using kernel point convolutions to capture local geometry, but TJYRoad-Net’s integration of MSNS allows it to handle larger, more complex urban scenes more effectively. Compared to SCF-Net, which achieves 94.23%, TJYRoad-Net’s hierarchical approach to context-aware feature aggregation enables more accurate segmentation, particularly in urban environments with densely packed structures.Traffic Poles and Small Objects: When it comes to small, sparse objects like traffic poles, TJYRoad-Net’s CAU provides a distinct advantage. With an IoU of 79.34%, it significantly outperforms RandLA-Net (71.43%) and SCF-Net (73.22%). Traffic poles pose a challenge due to their sparse representation in the point cloud and their vertical orientation, which can be confused with nearby structures. The RCE module enhances the network’s ability to recognize these small but important objects by encoding their spatial relations and orientation, ensuring that these features are not lost during the downsampling process. KPConv, while strong in capturing local geometry, sometimes struggles with extremely sparse objects, which explains its slightly lower IoU in this category (74.87%).Vegetation and Complex Natural Structures: For trees and other natural structures, TJYRoad-Net achieves an IoU of 97.07%, closely matching KPConv (95.23%) and SCF-Net (92.54%). Vegetation typically involves complex, irregular point distributions, which makes segmentation challenging. TJYRoad-Net handles this complexity by using the DWM to dynamically adjust the importance of different features based on their spatial relationships, which is crucial in distinguishing between overlapping objects like trees and buildings. This fine-grained control allows the network to segment trees and other complex natural structures with high accuracy, preserving details that are often missed by simpler models.

At lower segmentation precision, TJYRoad-Net shows certain limitations. For example, trees located near buildings are sometimes misclassified as buildings, likely due to the overlap in structural features captured in urban environments. Additionally, road edges are occasionally misclassified as green-belt areas, and certain areas near road markings exhibit jagged noise, which can reduce the overall segmentation clarity. These limitations highlight potential areas for improvement. Future work could explore refining the model to better distinguish overlapping or adjacent classes, particularly in complex urban scenes. Possible improvements include enhancing feature differentiation in boundary areas, reducing noise around high-contrast objects like road markings, and further optimizing the model’s handling of sparse, complex natural structures.

### 3.5. Fine Registration of Road Surface Points

The image point cloud contains nearly 20 times more ground points compared to the laser point cloud. Considering the complexity and computational cost during the ICP calculation process, the drone’s coarse-registered ground point cloud is downsampled to 1/20 of the initial point number using the VoxelGrid filter in the PCL library [37]. Subsequently, partial sections of both point clouds are selected for fine registration, which is shown in Figure 13. Figure 13 illustrates the error convergence curve observed during the iterative optimization process. The image point cloud, indicated by the blue section, represents the point cloud that requires registration. Conversely, the red section represents the target laser point cloud.

The error of the road surface point clouds from both sources converges to 3.002 cm and stabilizes at around 14 iterations. There is no further change in error after 20 iterations, and the iteration process is terminated at 24 iterations. Figure 14 presents the ultimate outcome of the registration process for the road surface point cloud.

### 3.6. Fine Registration of Non-Road Surface Points

To reduce the computational cost during the registration process, it is necessary to separately segment the main walls of the building façades from the image and laser point clouds. Since the TJYRoad-Net network has already extracted the complete building point clouds for both sources in Section 2, they can be directly processed. 

Among various methods for small-scale point cloud segmentation, region-growing-based segmentation methods can group point clouds with small differences together. The normal vectors and curvature information of building façade point clouds serve as suitable quantitative indicators for capturing these differences. Therefore, in this study, the region-growing segmentation method available in the PCL library is employed for the segmentation and extraction of building façades from the laser and image point clouds. The results are shown in Figure 15.

After segmentation, the building façade point clouds from both sources are finely registered using the ICP algorithm. The registration process and registration errors are illustrated in Figure 16.

The error of the building façade point clouds from both sources converges at around 16 iterations, and there is no further change in error after 18 iterations, with an error value of 2.492 cm. At this point, the registration process concludes. The rotation matrix R and translation vector t obtained from the building façade ICP solution are applied to the non-road surface point cloud captured by the drone. This registered point cloud is then concatenated with the road surface point cloud obtained in Figure 17. The final result of the fine registration is presented in Figure 17.

The registration accuracies of the two rounds of experiments are 3.002 cm and 2.492 cm, respectively. The errors fall within an acceptable range. This registration process effectively improves the overall accuracy of the image point cloud. With its advantages of wide coverage and high density, the image point cloud can be directly used for three-dimensional surface reconstruction of road infrastructure.

### 3.7. Road Surface Reconstruction

For straight road sections, the overall texture distribution is relatively uniform and the shape is regular. In contrast, the geometric characteristics of the road surface point cloud at intersections are more complex. This study focuses on reconstructing the road surface based on the intersection road segments.

The structure of the road surface in the scene mainly exhibits a striped distribution. After fusion, the density of the road surface point cloud can reach 8000 to 10,000 points per square meter. However, reconstructing a large-scale point cloud leads to an excessive number of triangular facets, which is not in line with the lightweight reconstruction goal of this study. The point cloud of road surface undergoes downsampling using the VoxelGrid filter available in the PCL library. Furthermore, different voxel sizes for downsampling the road surface model are investigated to explore the variation trend of the average model error. This provides a theoretical basis for selecting the appropriate voxel size for downsampling in road surface reconstruction. To evaluate the accuracy, the “Distance from Reference Mesh” function in Meshlab [38] is utilized. This function measures the geometric distance between the downsampled point cloud and the nearest neighboring triangular facet on the original model. It allows for a comparison of model errors. The variation in model error with downsampling voxel size is shown in Figure 18.

Based on the comprehensive analysis of the experimental results, when the voxel size is set to 11 cm or below, the model error of the road surface converges. When a voxel size of 12 cm is used, the average model error is reduced to 1.83 mm. For the specified voxel size, the road surface point cloud is expected to have a density of around 150 points per square meter. The RMSE of the downsampled point cloud is calculated to be 2.31 mm. Additionally, at a 95% confidence level, the maximum error does not surpass 6.3 mm. This meets the reconstruction accuracy and detail requirements for the intersection road surface, and the model satisfies the lightweight criteria. 

Figure 19 presents the road surface point cloud, showcasing the point cloud data both before and after downsampling. Meanwhile, Figure 20 showcases the outcome of reconstructing the road surface utilizing the Poisson reconstruction algorithm, employing a voxel size of 12 cm.

### 3.8. Template-Based Line Reconstruction

The direct Poisson reconstruction algorithm for road marking surface reconstruction produces the results shown in Figure 21. The problem with this approach is that the line mesh lacks overall regularity, exhibiting features such as jagged edges and inconsistent thickness. Some markings even appear incomplete. Using this model directly for subsequent applications would reduce the overall realism of the scene and contradict reality.

As road markings are highly regular road infrastructure, they must adhere to strict design specifications regarding size, shape, and other geometric structures. In this study, the road point cloud is subdivided, and this is followed by the utilization of a fitting spatial curve method to extract the road centerline. Furthermore, a standardized road marking template library is designed for the purposes of this study. The road markings are automatically recognized and generated based on the road centerline, replacing the original markings.

### 3.9. Template-Based Reconstruction of Road Markings

Since the elevation changes in the road are relatively gentle within a small range, the road centerline can be approximated by a three-dimensional polynomial curve model. The model can be represented as follows, where A, B, and C are the coefficients to be fitted:(8)$$z=A\left(x^{2}+y^{2}\right)+B\sqrt{x^{2}+y^{2}}+C$$

Curve fitting techniques for discrete point clouds are typically categorized into two categories: least squares and RANSAC. The least-squares fitting method uses all point cloud samples for curve fitting and is more robust compared to the RANSAC method, which randomly selects samples. It is more suitable for extracting road centerlines. Therefore, in this study, the least-squares method is used to determine the optimal solution for the coefficients A*, B*, and C* of the curve model equation.

First, the road point cloud is subdivided based on intersections, splitting the surveyed road network into multiple entrance and exit road segments. Curve fitting is performed on each segment’s point cloud. By comparing the larger differences in the x and y coordinates within each cluster of point clouds, the coordinate axis corresponding to the larger difference is selected as the grid subdivision reference axis. Each road segment’s point cloud area is subdivided into grids with a step size of 50 cm on the specific coordinate axis, and the centroid of each grid is calculated. The centroid obtained for segment 1 is shown in Figure 22, and the same process is applied to the remaining segments.

After obtaining the grid centroids for each road segment, the least-squares method is used with Equation (8) to perform curve fitting. The fitting results are shown in Table 7. By substituting the starting and ending points of each segment, the road centerline for each segment can be obtained.

### 3.10. Construction of the Template Library for Road Markings

In this study, the Profile Builder plugin in Sketchup Pro software is used to construct the template library for road markings. Based on the specified design shapes according to the regulations, parameters such as line segments, spacing length, and line width are set within the plugin. Road marking components are constructed as templates in the form of components, including lane dividers and pedestrian cross-walks. Additionally, various components are imported into an empty template in a modular manner to form a global template library for road markings, such as templates for bidirectional dual-lane or bidirectional four-lane roads. The framework of the template library is depicted in Figure 23.

Using the “Create Along Path” function in the Profile Builder plugin, road markings can be generated with a single click based on the extracted road centerline, matching the most suitable template. The generated results are shown in Figure 24.

### 3.11. Template-Based Reconstruction of Vegetation

Vegetation in road infrastructure includes trees, lawns, shrubs, etc. They have different volumes and shapes in the scene, and the point clouds corresponding to vegetation differ from those of other infrastructure elements, such as roads, markings, and buildings, which possess distinct and regular geometric features. Vegetation point clouds exhibit scattered, complex, and irregular characteristics, making it challenging to achieve ideal results by directly applying surface reconstruction algorithms to vegetation point clouds.

Although vegetation is abundant in the scene, the segmented vegetation point clouds exhibit a regular distribution with certain intervals between them. Therefore, in this study, geometric information such as size and position is extracted after individualizing the vegetation point clouds. A library of vegetation models for transportation scenes is constructed, and parameterized files are intelligently matched and substituted for the original point clouds.

During the individualization process of vegetation, the vegetation point clouds are first downsampled using a voxel filter (VoxelGrid). Subsequently, individualization is performed using a projection density clustering method following the algorithm steps below:

(a)Create a grid on the xOy plane based on the coordinate extremes of the original vegetation point cloud’s bounding box. Set the grid size (GridScale) to 10 cm in this study. Project all points onto each grid during traversal.(b)Create a 2D grid hash table (unordered_map) to record the index (Index) of each grid containing points and the count of points (count) in that grid.(c)Set a density threshold (T), which is set to 500 in this study. Extract the grids with a count greater than the threshold to form a new point cloud (P).(d)To perform spatial clustering of point clouds, employ the EuclideanClusterExtraction feature extractor from the PCL library, along with the creation of a KdTree. Adjust the clustering parameters as follows: Set the cluster tolerance to 1 m, specify the minimum cluster size as 100 points, and set the maximum cluster size to 250,000 points. For the individualization of vegetation, partition all vegetation point clouds into separate classes, with each class representing a distinct plant entity. The result of vegetation individualization is shown in Figure 25.

After individualization, each vegetation point cloud corresponds to its axis-aligned bounding box (AABB). The MomentOfInertiaEstimation feature extractor from the PCL library is used to extract the AABB of each vegetation, recording the maximum and minimum coordinates of each bounding box. This information is used to calculate the length, width, height, and bottom center coordinates of each bounding box, constructing a geometric database for vegetation assets. The extracted information is shown in Table 8.

This study utilizes Speedtree [39] to construct a library of plant models, which includes dozens of different sizes, colors, types, and styles of plant models, as shown. By intelligently matching the geometric information with the vegetation asset database, objects with the closest resemblance in appearance are selected and exported as 3D models in .obj or .st format. These models can be integrated into third-party rendering engines. The model library is depicted in Figure 25.

### 3.12. Generation and Application of Realistic Model

For other types of road infrastructure, this study employs different reconstruction methods. Similar to the approach used for road markings, a template-based replacement method is utilized for reconstructing poles. As for roadside buildings, a Poisson reconstruction method is applied. Finally, the road markings, road surfaces, and building models are imported into the UE4.27 rendering engine in .obj format. The vegetation and pole point clouds are imported as geometric replacement models. Additionally, the material textures of the models are optimized to enhance surface details and increase the realism of the scene. A comparison between the real and reconstructed scenes is depicted in Figure 26.

### 3.13. Application of Realistic Model

Upon completing the model generation process, although the scene closely resembles reality, as a traffic environment, the models still lack the physical logic of certain traffic participants and infrastructure elements. Therefore, this study further incorporates logic information for lanes, vehicles, traffic lights, and signs in the UE4.27 engine. Corresponding control classes were developed and applied to the traffic flow simulation and simulated driving modules.

#### 3.13.1. Traffic Flow Simulation Module

This study developed a subclass blueprint called “Car Source” by inheriting from the parent class “Wheeled Vehicle”. Before generating the traffic flow, various parameters need to be defined, such as the speed range of vehicles, the proportion of different vehicle sizes, the state of vehicle lights, the surface material of vehicles, the generation interval between vehicles, and the detection box range for object detection. Upon starting the drive, the object box range detection for obstacles is activated. The vehicles strictly follow the predefined lane blueprint for direction and randomly make decisions between going straight or turning based on the availability of the next lane at intersections. When obstacles are detected within the box range, the vehicles decelerate and avoid them based on preset deceleration rates. They come to a stop and wait if the obstacles have not disappeared yet. Similarly, when the vehicle enters the trigger box of a sign or traffic light, appropriate deceleration or stopping decisions are executed. The blueprint continuously monitors the vehicle’s coordinate position during the drive and is destroyed once it reaches the designated disappearance point. The simulation process is shown in Figure 26.

#### 3.13.2. Driving Simulation Module

Based on the realistic models built in UE4, this study has developed a driving simulation software platform. The platform is built on the underlying logic of the open-source plugin called AirSim, which is a vehicle and drone simulation platform based on UE. It allows vehicle control and the output of various types of simulated data, including current timestamps, XYZ absolute coordinates, brake-throttle-steering inputs, RPM, etc., using corresponding scripts at a certain sampling rate. AirSim provides multiple API interfaces for secondary development.

In this study, the native appearance and control models of vehicles in AirSim were redesigned. The camera position and orientation inside the vehicle were adjusted to a suitable first-person perspective for the driver. Headlights and taillights were added with adjustable brightness. The engine sound amplitude was set based on different RPM values. The redesigned models inherit the main control vehicle’s “SUV Car Pawn” class, and the configuration of cameras and vehicle models was performed in the Settings.json file. At the same time, the interaction logic between the main control vehicle’s “SUV Car Pawn” class and the environmental vehicle’s “Car Source” class was implemented. This allows the driver to interact with the environmental vehicles in real-time. The driving simulation interface is shown in Figure 27.

This study formats and stores the driving simulation data, and develops an easy-to-use, dynamic, and interactive data visualization and analysis platform based on C++ and QT Creator 5.10.0. This platform provides efficient and lightweight data querying and display functions for user subjects, as shown in Figure 27.

## 4. Conclusions

This paper presents a comprehensive approach to digitizing transportation infrastructure, covering data acquisition, object classification, data fusion, model reconstruction, and practical applications. These contributions support intelligent transportation and digital twin applications. Based on the comparative analysis of the segmentation experimental results for laser point clouds and image point clouds, three key conclusions can be drawn:

(a)Enhanced Feature Utilization for Segmentation: Road infrastructure point clouds exhibit distinct characteristics—smoothness and continuity in road markings, planar road surfaces, and verticality in poles and signs. The RCE and DWM modules in TJYRoad-Net effectively leverage these features, enhancing segmentation accuracy. The integration of normal vector information in the RCE was essential for distinguishing road surfaces from buildings and other elements.(b)Optimized Transfer Learning: Selectively fine-tuning the fourth decoder layer in TJYRoad-Net resulted in the best performance, achieving an mIoU of 90.77% and OA of 97.01% for smaller target datasets. This method improved generalization while preventing overfitting. Compared to state-of-the-art models, TJYRoad-Net achieved a 2–5% improvement in mIoU, highlighting its robustness and adaptability.(c)Precise Registration and Lightweight Reconstruction: Initial registration with RANSAC followed by the ICP achieved a precise 2–3 cm alignment accuracy between image and laser point clouds. For lightweight surface reconstruction, a point cloud density of 150 points/m^2^ produced average elevation errors of 1.83 mm, within a 6.3 mm confidence interval. Centerline fitting and template matching for road markings, along with bounding boxes for plants, resulted in accurate and efficient models.

The realistic models generated in this study can be further applied in 3D modeling platforms to enable visualization, virtual roaming, simulated driving, and traffic simulation functionalities. Overall, this research contributes to the digitization of transportation infrastructure and provides a foundation for the development of visualization and simulation applications in the fields of intelligent transportation and digital twins.

## Figures and Tables

**Figure 1 sensors-24-07222-f001:**
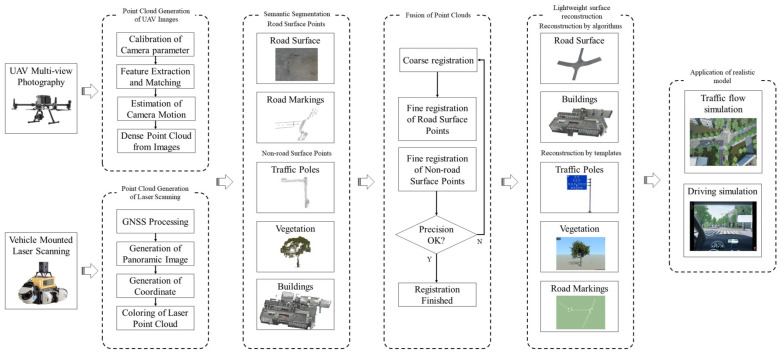
The technical roadmap of the entire paper.

**Figure 2 sensors-24-07222-f002:**
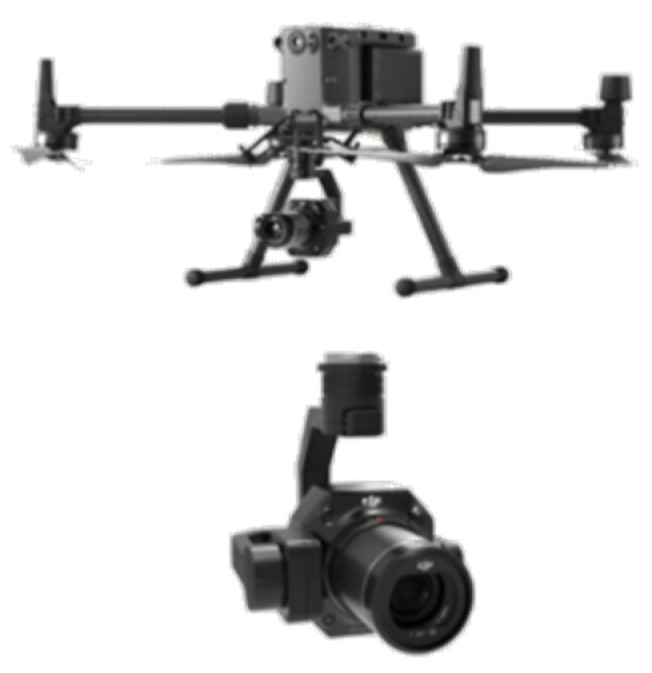
DJI M300RTK UAV with Zenith P1 gimbal camera.

**Figure 3 sensors-24-07222-f003:**
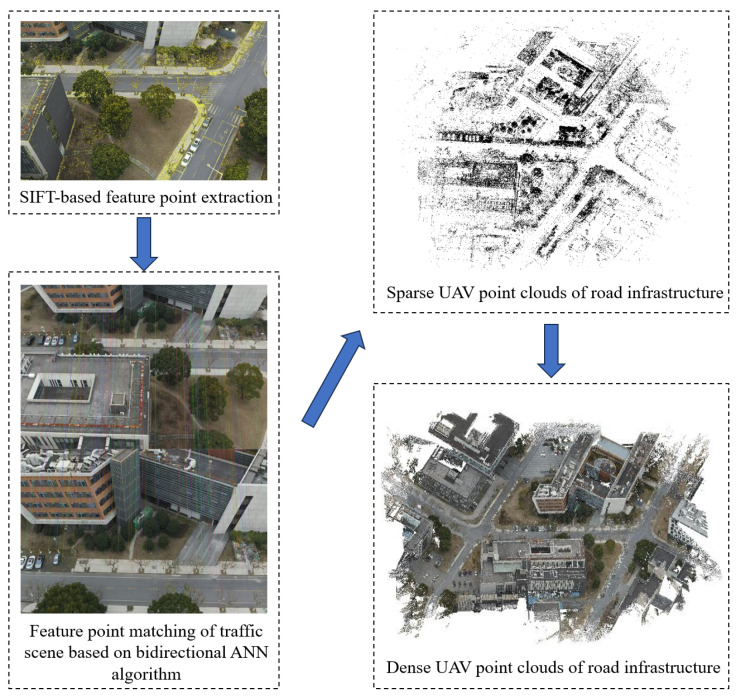
Dense UAV point cloud of road infrastructure.

**Figure 4 sensors-24-07222-f004:**
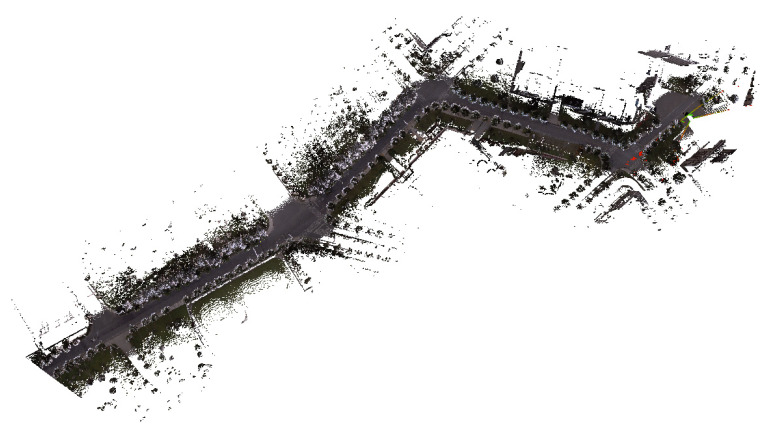
Laser point cloud of road infrastructure.

**Figure 5 sensors-24-07222-f005:**
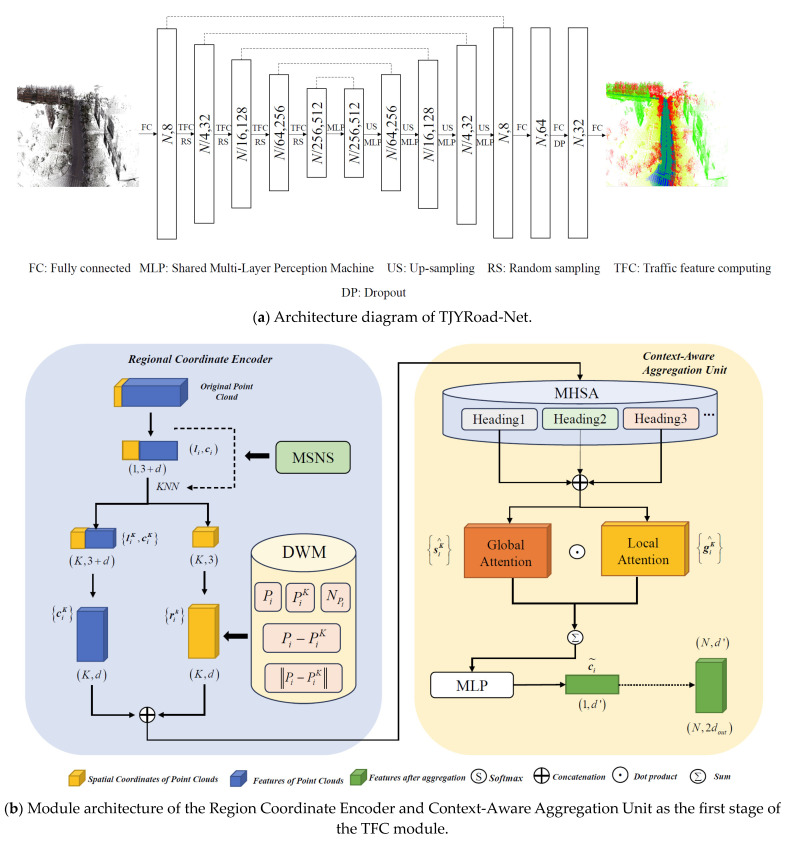
TJYRoad-Net network.

**Figure 6 sensors-24-07222-f006:**
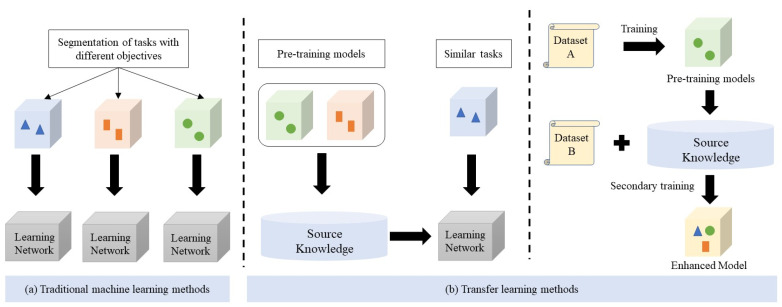
Traditional machine learning versus transfer learning.

**Figure 7 sensors-24-07222-f007:**
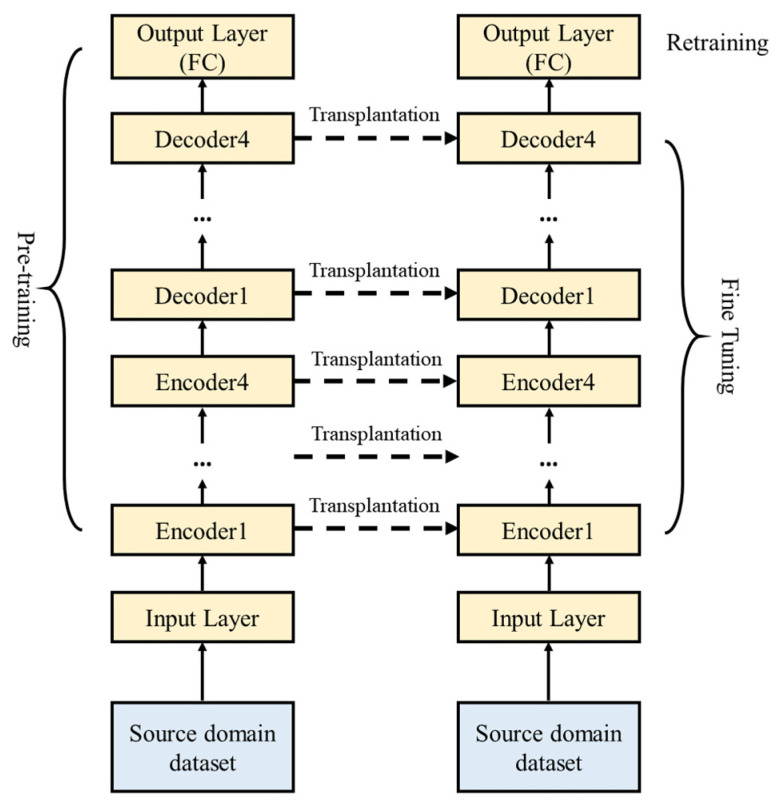
Fine-tuning ideas of enhanced TJYRoad-Net.

**Figure 8 sensors-24-07222-f008:**
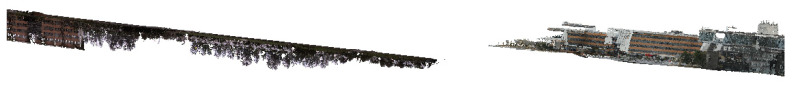
Image point cloud and laser point cloud.

**Figure 9 sensors-24-07222-f009:**
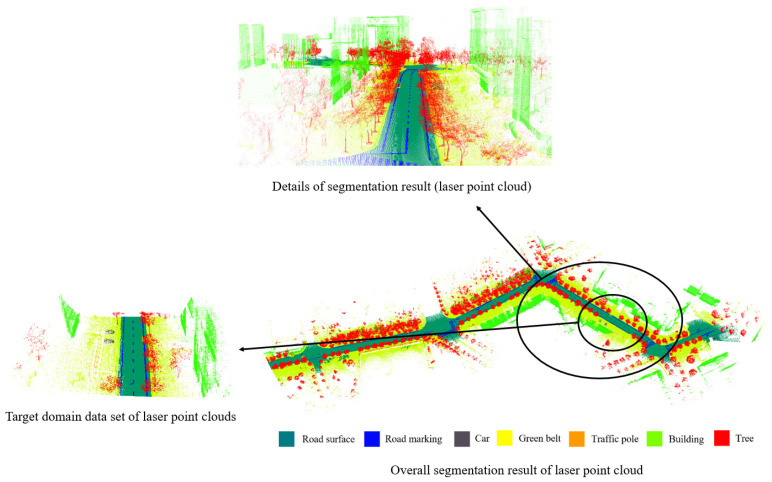
Semantic segmentation result of laser point cloud.

**Figure 10 sensors-24-07222-f010:**
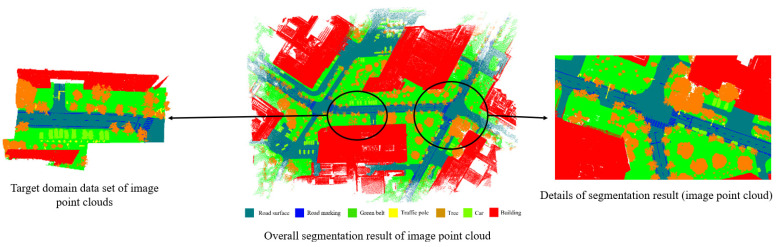
Semantic segmentation results of image point cloud.

**Figure 11 sensors-24-07222-f011:**
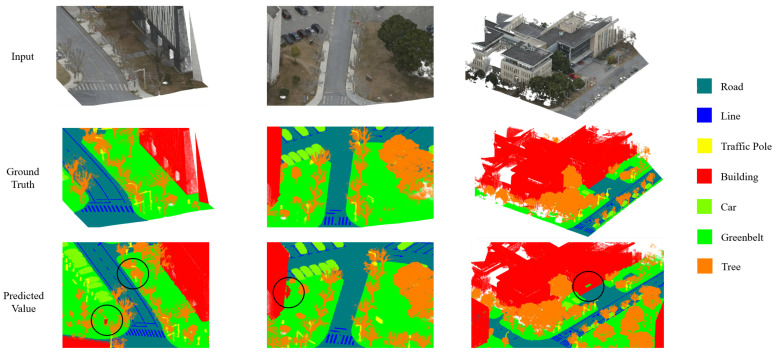
Semantic segmentation results of image point clouds from a road intersection scene, showing Input (original point cloud), Ground Truth (manually annotated labels), and Predicted Value (model output with misclassifications circled).

**Figure 12 sensors-24-07222-f012:**
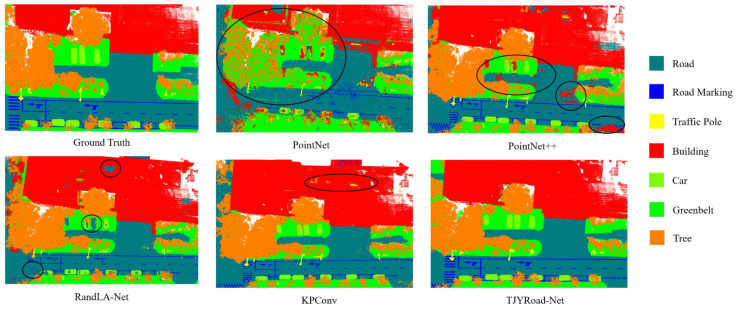
Comparison of segmentation results across different state-of-the-art methods, with red circles highlighting the segmentation outputs at identical locations for each method.

**Figure 13 sensors-24-07222-f013:**
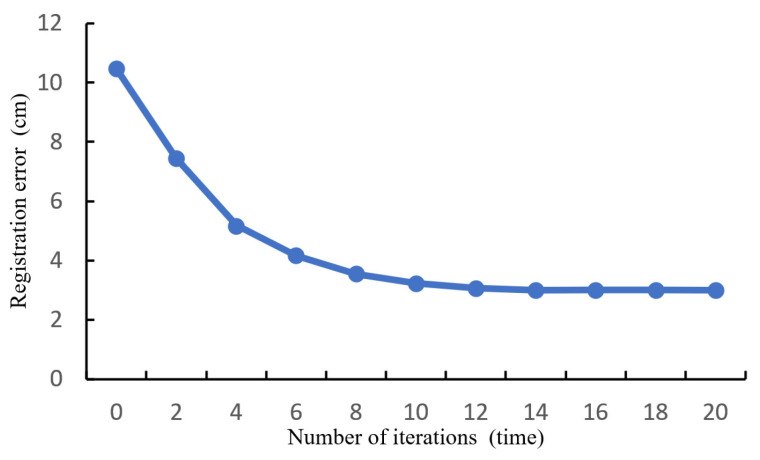
ICP fine alignment error of pavement point cloud.

**Figure 14 sensors-24-07222-f014:**
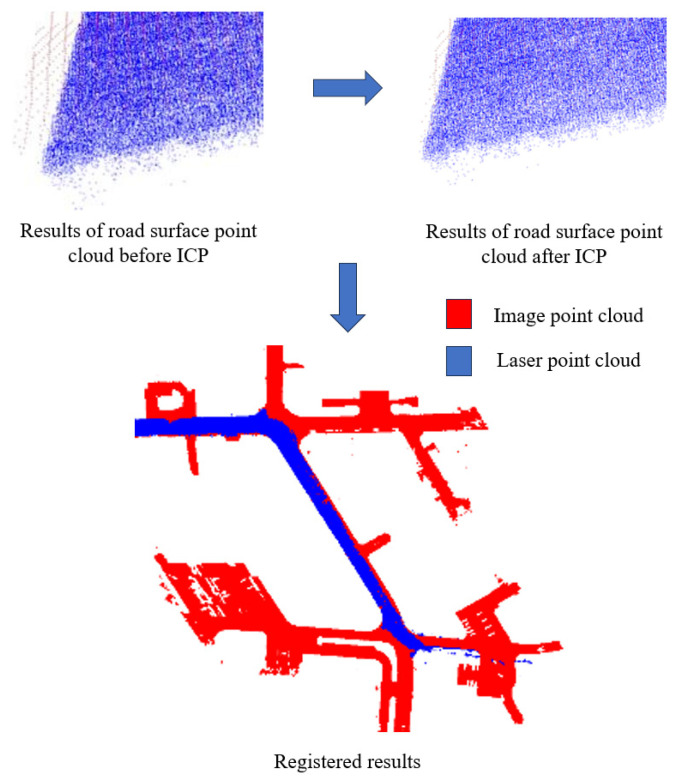
Registered results of road surface point clouds.

**Figure 15 sensors-24-07222-f015:**
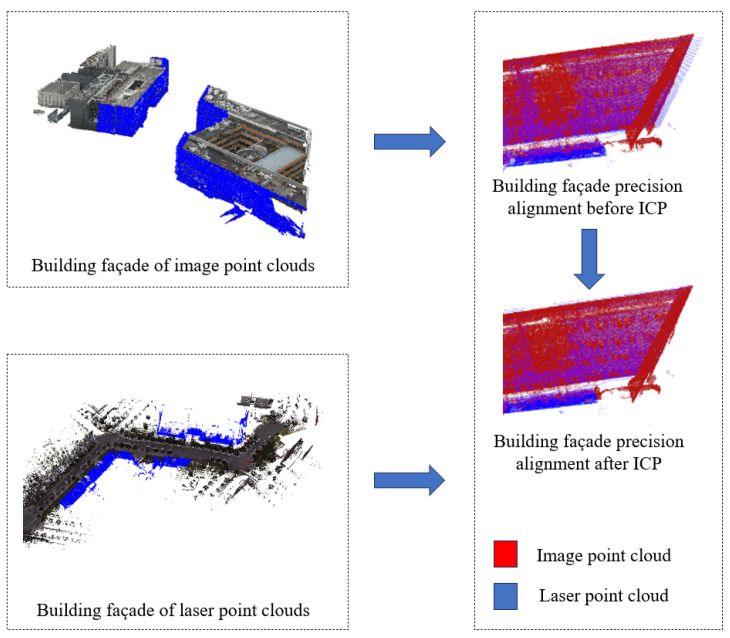
Process of building façade precision.

**Figure 16 sensors-24-07222-f016:**
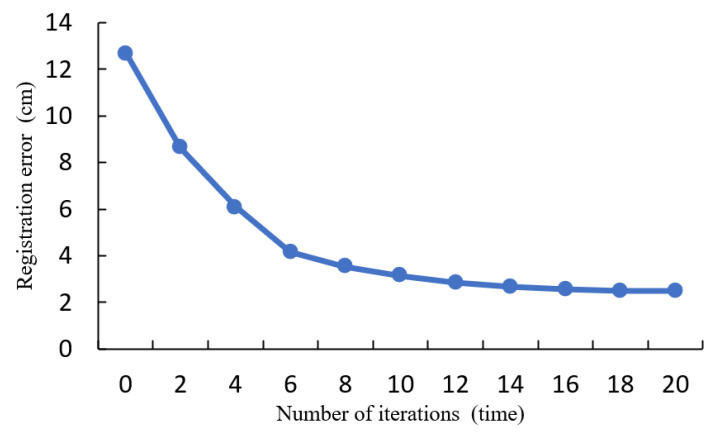
ICP fine alignment error of building façade point clouds.

**Figure 17 sensors-24-07222-f017:**
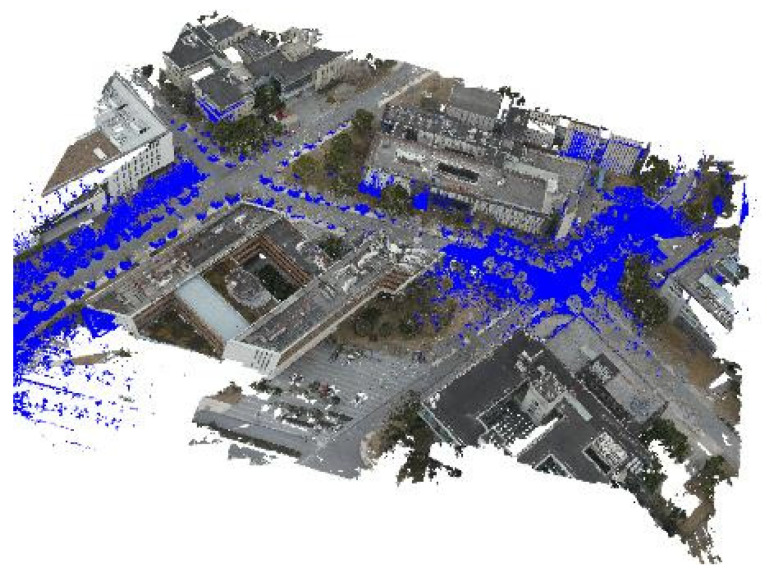
Alignment result of point clouds.

**Figure 18 sensors-24-07222-f018:**
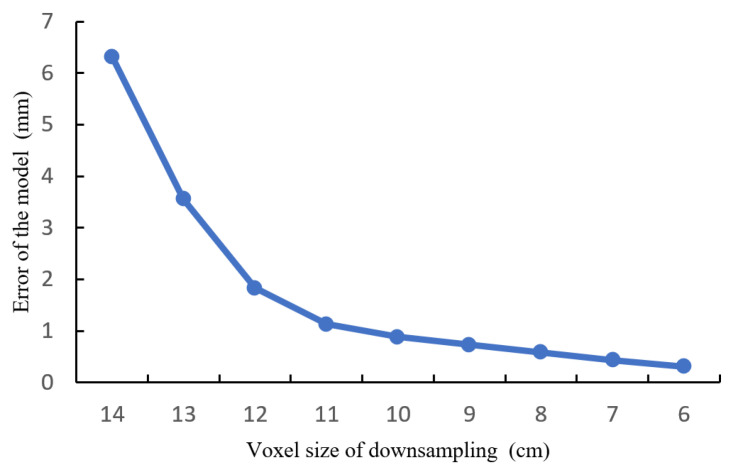
Variation in model error with downsampling voxel size.

**Figure 19 sensors-24-07222-f019:**
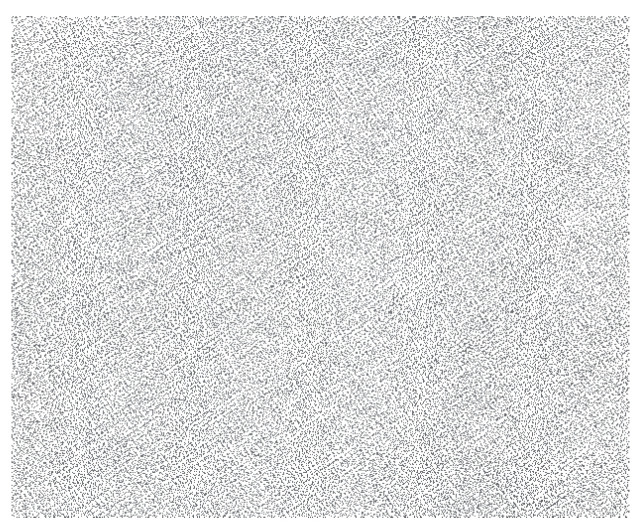
Downsampling results of road surface point clouds.

**Figure 20 sensors-24-07222-f020:**
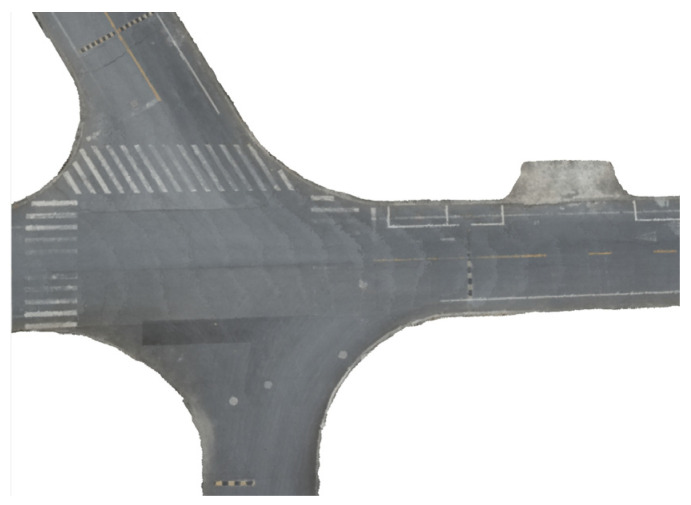
Result of road reconstruction.

**Figure 21 sensors-24-07222-f021:**
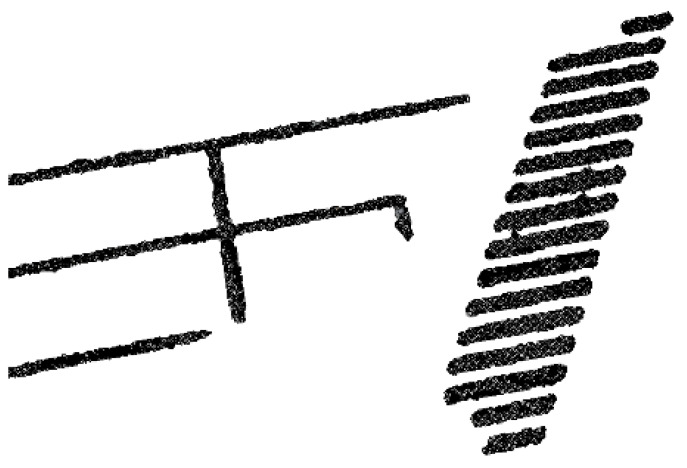
Marker triangle network structure based on Poisson reconstruction.

**Figure 22 sensors-24-07222-f022:**
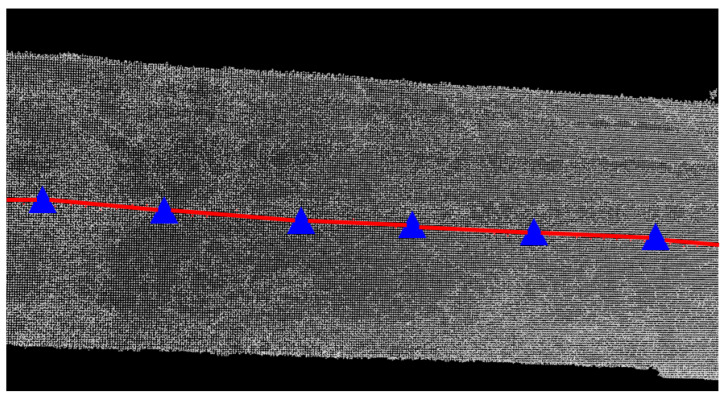
A section of the grid center of mass.

**Figure 23 sensors-24-07222-f023:**
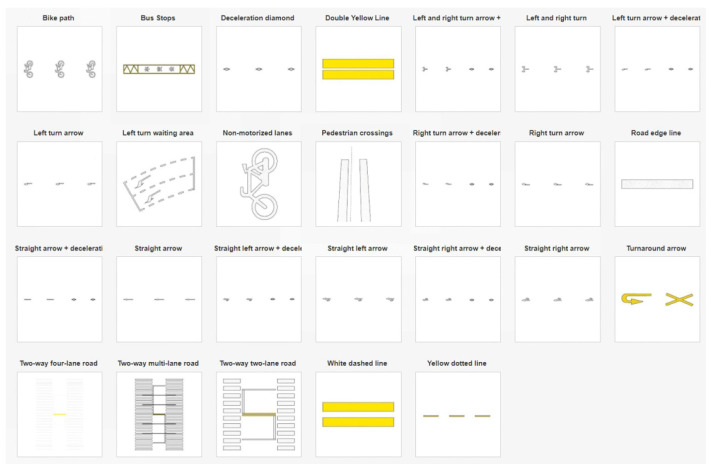
Design of road marking template library.

**Figure 24 sensors-24-07222-f024:**
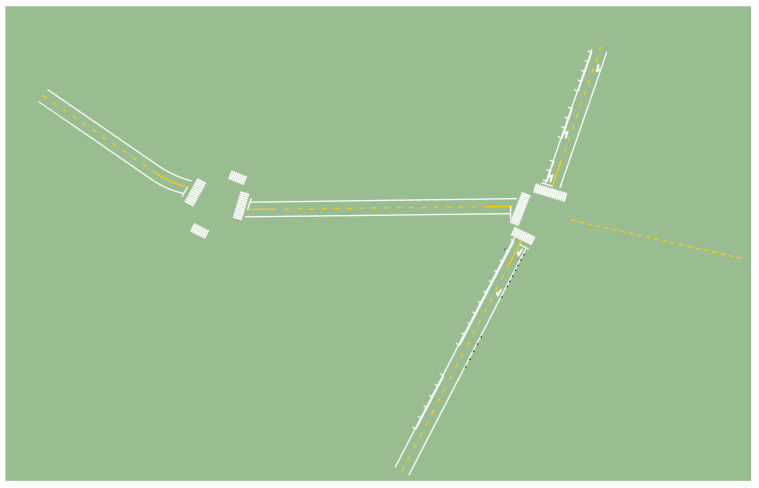
Marking reconstruction results.

**Figure 25 sensors-24-07222-f025:**
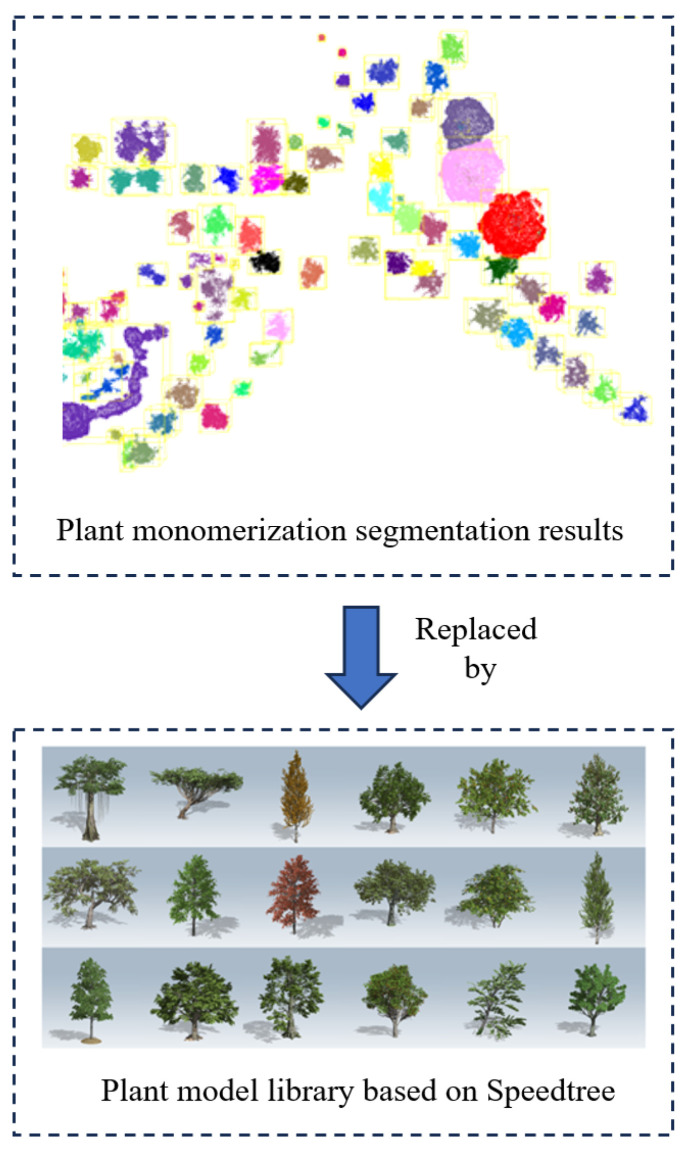
Vegetation reconstruction results.

**Figure 26 sensors-24-07222-f026:**
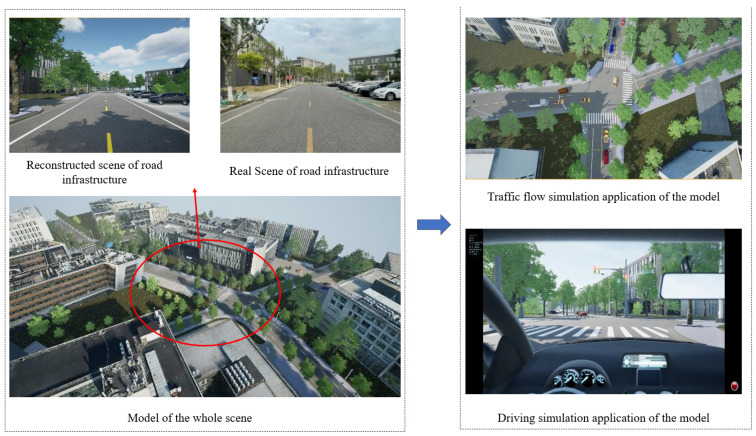
Real scene of road infrastructure.

**Figure 27 sensors-24-07222-f027:**
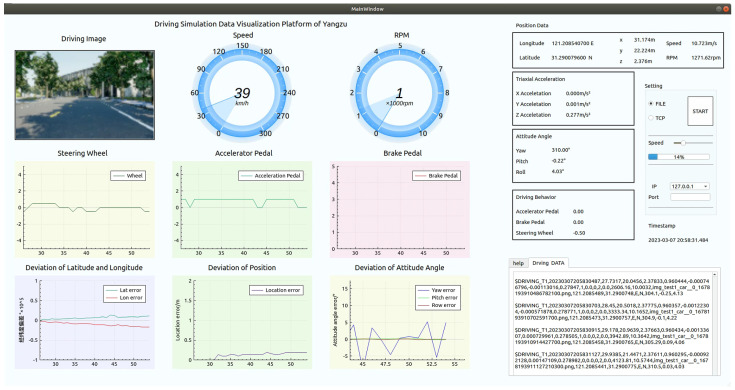
Driving simulation data visualization platform.

**Table 1 sensors-24-07222-t001:** Technical specifications for UAV data acquisition.

Parameter	Specification
UAV Model	DJI Matrice M300 RTK
Camera Model	Zenmuse P1
Sensor Type	Full-frame
Focal Length	35 mm F2.8 LS ASPH
Image Resolution	45 million pixels (8192 × 5460)
Pixel Size	4.4 µm
Plane Accuracy (RTK)	±3 cm
Elevation Accuracy (RTK)	±5 cm
Field of View (FOV)	63.5°
Ground Sampling Distance (GSD)	0.75 cm/pixel (at 60 m altitude)
Altitude (Flight Height)	60 m
Survey Area	21,810 m^2^
Longitudinal Overlap	80%
Lateral Overlap	70%

**Table 2 sensors-24-07222-t002:** Technical specifications for LiDAR data acquisition.

Parameter	Specification
System Model	Topcon IP-S2
Laser Scanners	3 × SICK laser scanners
Camera	6 × LADYBUG panoramic CCD cameras
Positioning System	Dual-frequency GPS GLONASS
Inertial Navigation Unit	Included
Wheel Encoders	2 units
Laser Measurement Accuracy	±35 mm
Laser Range	Up to 30 m
Camera Resolution	1600 × 1200 pixels
GNSS Dynamic Measurement Accuracy	10–15 mm
Total Survey Distance	1.8 km

**Table 3 sensors-24-07222-t003:** Experimental hyperparameter settings for point cloud segmentation.

Groups	Experimental Setup	Improves the Network or Not	Batch_Size	Learning Rate	Input Points	Training Rounds
1	Direct training	No	4	0.01	65,536	20
2	Fine-tune all layers	No	6	0.001	40,960	15
3	Fine-tune output layer	No	6	0.001	40,960	15
4	Fine-tune decoder 4	No	6	0.001	40,960	15
5	Fine-tune decoder 3 and 4	No	6	0.001	40,960	15
6	Direct training	Yes	4	0.01	65,536	20
7	Fine-tune all layers	Yes	6	0.001	40,960	15
8	Fine-tune output layer	Yes	6	0.001	40,960	15
9	Fine-tune decoder 4	Yes	6	0.001	40,960	15
10	Fine-tune decoder 3 and 4	Yes	6	0.001	40,960	15

**Table 4 sensors-24-07222-t004:** Experimental results of laser point cloud segmentation.

Groups	IoU (%)							mIoU (%)	OA (%)
	Road Marking	Road Surface	Green Belt	Car	Traffic Pole	Tree	Building		
1	1.43	42.84	34.83	49.34	7.82	42.96	12.28	27.36	51.81
2	31.67	86.61	71.36	82.63	46.35	76.67	81.33	68.09	88.93
3	15.98	78.59	58.18	71.85	24.41	68.40	74.73	66.28	88.17
4	47.83	92.03	76.69	91.67	67.33	81.94	85.69	77.60	93.12
5	41.67	85.93	77.38	89.64	51.18	83.33	79.76	72.70	91.38
6	12.67	63.28	41.29	63.17	26.48	51.69	42.89	43.07	75.38
7	48.36	91.33	82.87	89.41	82.11	93.71	81.52	81.33	93.48
8	43.76	86.24	76.28	92.64	79.67	91.16	76.89	78.09	92.54
9	56.38	93.94	89.64	95.52	81.16	95.28	90.06	86.00	95.12
10	51.14	83.33	91.02	93.28	76.51	92.46	88.64	82.34	93.95

**Table 5 sensors-24-07222-t005:** Experimental results of image point cloud segmentation.

Groups	IoU (%)							mIoU (%)	OA (%)
	Road Marking	Road Surface	Green Belt	Car	Traffic Pole	Tree	Building		
1	2.01	71.37	63.02	64.87	21.6	68.16	29.58	45.80	76.33
2	27.94	82.26	75.88	74.04	57.61	83.71	91.83	70.47	91.17
3	16.96	73.05	78.48	82.34	55.71	91.48	92.22	70.03	91.05
4	42.68	85.62	86.13	79.28	67.12	93.41	94.27	78.35	93.43
5	34.11	79.41	79.21	86.26	55.54	86.26	96.33	73.87	91.86
6	10.36	69.55	61.93	58.72	24.93	74.31	56.83	50.95	78.38
7	54.54	92.34	95.07	92.02	82.98	98.13	99.02	87.73	95.16
8	51.28	85.68	71.26	76.67	62.64	86.93	51.16	69.37	90.86
9	79.28	95.34	94.24	92.87	79.34	97.07	97.22	90.77	97.01
10	54.50	92.36	95.28	92.46	83.54	98.22	99.11	87.92	95.97

**Table 6 sensors-24-07222-t006:** Comparison with state-of-the-art methods.

Methods	IoU (%)							mIoU (%)	OA (%)
	Road Marking	Road Surface	Green Belt	Car	Traffic Pole	Tree	Building		
PointNet [17]	35.12	55.43	50.78	48.32	30.74	60.23	55.78	48.06	65.23
PointNet++ [18]	40.45	65.89	65.12	62.23	40.78	70.45	65.78	58.67	70.45
RandLA-Net [15]	73.43	90.12	88.23	89.22	71.43	91.87	93.34	85.38	91.56
SCF-Net [16]	74.32	91.45	89.78	90.67	73.22	92.54	94.23	86.60	92.78
KPConv [36]	77.12	93.87	92.43	91.23	74.87	95.23	96.78	88.79	94.67
TJYRoad-Net	79.28	95.34	94.24	92.87	79.34	97.07	97.22	90.77	97.01

**Table 7 sensors-24-07222-t007:** Optimal solution of curve fitting for each road section.

Number of Road	A*	B*	C*	Start Point	End Point
1	0.000125	−0.038196	26.0908	(−106.09, 18.84, 3.58)	(−47.06, −15.94, 3.68)
2	0.000018	0.017523	24.1397	(−83.32, −105.59, 4.00)	(−41.17, −33.87, 4.17)
3	0.000105	−0.044156	24.7202	(28.55, 72.39, 4.05)	(−21.53, −4.19, 4.55)
4	−0.000042	0.001244	24.9557	(−15.54, −23.45, 4.07)	(103.97, −20.94, 4.80)
5	−0.000027	−0.03464	28.8755	(156.61, 46.98, 4.72)	(124.54, −11.88, 4.40)
6	−0.002946	0.83765	−31.8425	(65.48, −122.57, 4.60)	(110.57, −37.74, 4.28)

A*, B*, and C*: Coefficients determined by the least-squares method to achieve the optimal solution for the curve model equation.

**Table 8 sensors-24-07222-t008:** Asset database of plants.

Number of Boxes	Length	Weight	Height	Center Coordinates of Bottom Surface
1	7.02	7.02	12.26	(14.24, 15.82, 4.92)
2	6.18	6.17	7.51	(−49.62, −36.28, 4.56)
3	7.23	9.41	7.76	(−9.06, −18.86, 4.53)
4	6.59	5.84	8.03	(3.16, 24.70, 4.66)
5	7.47	6.16	8.20	(93.30, −15.38, 4.75)
6	4.81	4.74	7.98	(−4.90, 11.76, 4.88)
7	5.29	6.45	6.56	(−4.54, −29.28, 4.18)
8	3.29	3.56	2.66	(−26.80, −36.60, 4.63)
…	…	…	…	…

## Data Availability

Data are contained within this article.

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
