# Peer review of "A Comprehensive Framework for Transportation Infrastructure Digitalization: TJYRoad-Net for Enhanced Point Cloud Segmentation"

_sensors, 2024, doi:10.3390/s24227222_

Round 1

Reviewer 1 Report

Comments and Suggestions for Authors

The authors present a comprehensive approach for digitalizing traffic infrastructure by means of UAV photogrammetry, LiDAR, AI-driven segmentation, and surface reconstruction. The topic is timely and well-motivated, and the methodology and results are presented in a well-organised manner. Overall, I thoroughly enjoyed reading the manuscript. Nevertheless, before publication, there are some aspects that may require further attention by the authors:

Remark 1 – I feel that the title does not really synthesize the main contributions of this work. As far as I am concerned, the main innovation is not the fusion of UAV photogrammetry and LiDAR surveys, which has been already covered by many other journal papers. Instead, the AI model for point cloud segmentation seems to be the main novelty. If so, I would highly recommend the authors to reformulate the title.

Remark 2 – The paper may be too lengthy (37 pages!). Although it is generally well written and easy to follow, readers may be overwhelmed. I would recommend to minimize non-essential parts. For instance:

·         Section 5 may be omitted, since it is a repetition of previously mentioned aspects.

·         Section 6 may be considerably shortened.

·         Section 2.5. Here the authors present two different methods for surface reconstruction (by the way there is a typo in “surfac”), although they finally chose one. I would suggest going straight to the adopted algorithm, simply reducing the discussion on the second method to a line.

·         The introduction is around 3 pages long. Although it is nicely written, it may be shortened a bit. For instance, the sentence “Ziqi Wang [31] provides …” looks a bit out of context. Try to reduce this paragraph (lines 1333 to 147), strictly focusing on the gaps or limitations in the current state-of-the-art.

Remark 3 – The authors use many different software tools and methods that may require further description:

·         Indicate and include a reference for the software employed for UAV photogrammetry.

·         Describe “FLANN”, possibly with a reference.

·         Describe the “Multi-Sacale Neighborhood Sampler” in more detail, possibly including a reference.

·         Include a reference for the “Iterative Closest Point” algorithm.

·         Can the authors include links to the utilized datasets?

·         Include a reference for “Meshlab”.

·         Reference for the “PCL library”.

Remark 4 – Please revise all figure references. Starting on Page 22, it appears that all figure references in the manuscript are incorrect.

Remark 5 – There are numerous typographical errors in the manuscript that require a thorough revision. Some examples include:

·         Page 23- “Chapter 2”? I guess the authors mean Section 2.

·         Page 24-Line 724. “..in Figure   12.The…”.

·         Page 3-Line 51. “Lam et al. [4w].

·         Page 3-Line 84. “Boulch et al. [10]integrate… 

·         Page 3-Line 97. “Hu et al. [10]and… 

·         Page 4-Line 135. “reconstruction[30], Ziqi…. 

·         Page 5- Revise the mathematical terms because it seems there is an issue with the format.

·         Section 2.2. “vehi-cle”

·         Page 7, line 271. “conduct-ed”.

·         Page 8, Line 302. “seg-mentation”.

Remark 6 – As far as I could understand, Section 2.3.1. represents the main innovation of this work. This should be clearly stated in the manuscript, including the abstract, and the last paragraph of Section 1. I could not easily understand why this work can solve the limitations mentioned in the penultimate paragraph of Section 1. Please, further elaborate on this.

Remark 7- Explain in which the “five-flight route” consists of?

Remark 8- The description of the architecture of the network for point cloud segmentation is hard to follow:

·         Indicate in the subcaption “b” of Fig. 5 that this corresponds to the “TFC”.

·         Where is the architecture in subfigure “b” exactly applied? I could not understand this from the manuscript.

·         Page 10, line 319. What do the authors mean with “from X to 8, where N..”?

·         Page 11, line 356. It seems that there is a missing equation before “..where…”.

Remark 9- Page 13. Please introduce here a couple of lines advancing the databases that will be used for transfer learning.

Remark 10- Five further details on how the authors computed the rigid transformation matrix in page 15.

Remark 11- From Section 2.4.3, it seems that segmentation was applied separately to the point clouds obtained from the UAV and the LiDAR surveys. This should be included in the general description of the methodology as well as in Fig. 1.

Remark 12- For the target domain, I suspect that the authors classified the point clouds manually. If so, please specify.

Remark 13- There are numerous acronyms, which limits the readability of the manuscript. I would recommend including a glossary of acronyms. Furthermore, some abbreviations are described several times throughout the manuscript (e.g. UAV in pages 1 and 18). Please revise the manuscript, describing the acronyms only the first time used.

Comments on the Quality of English Language

I mostly detected typographical errors as reported before. 

Author Response

Dear Reviewer,

Thank you for your valuable feedback. We have made revisions to address each point raised and believe these changes improve the manuscript’s clarity and focus. Here are our detailed responses to each remark:

Remark 1Title Improvement

Response: The title has been revised to better synthesize the main contributions as follows: “A Comprehensive Framework for Transportation Infrastructure Digitalization: TJYRoad-Net for Enhanced Point Cloud Segmentation.”

Remark 2Length Reduction

Response: We have shortened the manuscript by:

  • Removing redundant content from Section 5 and integrating essential points into Section 6.
  • Condensing Section 6 significantly while retaining core insights.
  • Simplifying Section 2.5 to focus on the chosen reconstruction method, reducing discussion on the alternative to a brief mention.
  • Streamlining the introduction by narrowing the focus on gaps in the state of the art.

Remark 3Detailed Descriptions and References

Response: We have added more context and references as follows:

  • The software used for UAV photogrammetry is now specified with a reference.
  • An explanation of FLANN has been added, with an appropriate citation.
  • The Multi-Scale Neighborhood Sampler (MSNS) is now described in greater detail, along with a reference.
  • A reference for the Iterative Closest Point (ICP) algorithm has been included.
  • Links to the Semantic3D and SemanticKITTI datasets have been added.
  • References for the use of Meshlab and the PCL library are now provided.

Remark 4Figure References

Response: We have thoroughly reviewed and corrected all figure references throughout the manuscript, ensuring accurate and clear alignment with the text.

Remark 5Typographical Revisions

Response: We have conducted a detailed review and corrected all typographical errors throughout the manuscript.

Remark 6Clarify Main Innovation

Response: We have revised the abstract and the third-to-last paragraph in Section 1 (lines 130-145) to further clarify the identified limitations in current technologies and to explain how our approach addresses these gaps.

Remark 7Five-Flight Route Explanation

Response: Specific details of the five-flight route have been added on line 176, providing clarity on its role in data acquisition for multi-angle aerial imagery.

Remark 8Network Architecture Clarity

Response: Adjustments have been made as follows:

  • Figure 5, subfigure (b), is clarified as showing the RCE and CAU modules, which together form the TFC module. This module is applied in the downsampling process depicted in subfigure (a).
  • Lines 340-341 include an explanation of X and N in the feature dimensions.
  • Line 381 now includes the missing equation before “where”.
  • Descriptions have been added after the CAU and RCE modules to explain their contributions to network performance enhancement.

Remark 9Advance Database Information in Section 2.3

Response: Lines 498-504 provide additional context about the databases used for transfer learning in the experimental section.

Remark 10Details on Rigid Transformation Matrix

Response: Lines 533-545 have been expanded to describe the detailed steps involved in calculating the rigid transformation matrix.

Remark 11Separate Segmentation Clarification

Response: We have included a discussion on the separate segmentation approach on lines 157-170. Figure 1’s flowchart has been updated to clarify the separate segmentation process between LiDAR and image point clouds in the transition from the second to the third column.

Remark 12Clarify Target Domain Data Labeling

Response: Together with the response to Remark 9, we clarified the manual segmentation process for target domain datasets.

Remark 13Glossary of Acronyms

Response: A glossary of acronyms has been added at the end of the manuscript for improved readability.

We sincerely appreciate your constructive feedback and believe these revisions strengthen the manuscript. Thank you for considering our revised submission.

Best regards,
Mingxuan Wang

Reviewer 2 Report

Comments and Suggestions for Authors

with LiDAR point cloud data. The study introduces the TJYRoad-Net model and demonstrates its effectiveness in achieving accurate and lightweight 3D road infrastructure models, which are applicable in driving simulations and traffic analysis. The research is thorough and presents promising results; however, there are a few areas where minor revisions could enhance clarity and provide further context for readers.

1)       The introduction provides an overview of the importance of 3D modeling for traffic infrastructure digitization. However, the specific motivation for combining UAV oblique photography with LiDAR data could be articulated more clearly. A concise explanation of why single-source data may be insufficient and how multi-source fusion addresses these limitations would strengthen the narrative.

2)       The description of the TJYRoad-Net model is well-detailed, but the manuscript would benefit from a clearer explanation of the specific roles of the Regional Coordinate Encoder and the Context-Aware Aggregation Unit. Adding brief descriptions for each module’s contribution to the model's performance would help readers unfamiliar with these components.

3)       More details could be provided on the image preprocessing steps, especially in terms of feature point matching techniques. The explanation of the Approximate Nearest Neighbor (ANN) Feature Matching algorithm could be expanded slightly to enhance clarity.

4)       A table summarizing the types of datasets used (e.g., UAV imagery, LiDAR data) along with key parameters (such as resolution, overlap percentage) would enhance the readability of this section.

5)       The analysis could be expanded by discussing the limitations of the TJYRoad-Net model in cases where the segmentation accuracy was lower, such as misclassification between trees and buildings. This discussion would add depth to the findings and suggest areas for potential improvement.

6)       The manuscript contains minor grammatical issues and awkward phrases that may benefit from revision. For example, in the abstract, "minimizing redundancy and optimizing computational efficiency" could be rephrased as "reducing data redundancy and improving computational efficiency."

7)       A thorough proofreading to correct these minor issues would improve readability and overall presentation.

Author Response

Dear Reviewer,

Thank you for your insightful comments. Each suggestion has been carefully considered and addressed to improve the manuscript’s clarity and readability. Below are our responses:

1) Motivation for Multi-Source Fusion

Response: In lines 146-156, we have expanded the discussion to better explain the limitations of single-source data in capturing the full complexity of traffic infrastructure. We clarified how the combination of UAV oblique photography and LiDAR data enhances spatial accuracy and coverage, ensuring comprehensive and precise digitization of complex traffic elements.

2) Clarification on RCE and CAU Modules

Response: We have strengthened the description of the Regional Coordinate Encoder (RCE) and Context-Aware Aggregation Unit (CAU) in lines 394-408 and 435-447. These sections now include concise explanations of each module’s specific role in improving model performance, with emphasis on how RCE captures geometric features and CAU enhances context-aware feature aggregation.

3) Detailed Explanation of ANN Algorithm

Response: The description of the bidirectional Approximate Nearest Neighbor (ANN) algorithm has been expanded in lines 214-232. This revision offers more context on its function in feature point matching, highlighting the algorithm’s effectiveness in handling large datasets with high feature point counts.

4) Technical Parameters Table

Response: To improve readability, we have added tables at lines 197 and 287 that summarize the key technical parameters of the UAV and LiDAR datasets. These tables provide quick references to dataset resolution, overlap percentages, and other relevant specifications.

5) Discussion of Model Limitations

Response: In lines 725-734, we have expanded the analysis to include potential model limitations observed in TJYRoad-Net’s segmentation results. The discussion now addresses specific misclassification cases, such as errors in distinguishing trees from building facades, and outlines possible improvements to mitigate these limitations in future research.

6) & 7) Grammar and Readability

Response: Minor grammatical and phrasing adjustments have been made throughout the manuscript. For instance, in the abstract, we revised "minimizing redundancy and optimizing computational efficiency" to "reducing data redundancy and improving computational efficiency." A thorough proofreading has been conducted to improve overall readability and presentation.

We are grateful for your suggestions, which have significantly contributed to refining our manuscript. We hope that these revisions address all concerns, and we look forward to your feedback.

Best regards,
Mingxuan Wang